# Resident microbial communities inhibit growth and antibiotic-resistance evolution of *Escherichia coli* in human gut microbiome samples

Michael Baumgartner[1]*, Florian Bayer[2], Katia R. Pfrunder-Cardozo[1], Angus Buckling[2], Alex R. Hall[1]

**1** Institute of Integrative Biology, Department of Environmental Systems Science, ETH Zürich, Zürich, Switzerland, **2** Biosciences, University of Exeter, Penryn, Cornwall, United Kingdom

* michael.baumgartner@env.ethz.ch

**Data Availability Statement:** Sequences have been deposited in the European Nucleotide Archive under the study accession number PRJEB36309

## Abstract

Countering the rise of antibiotic-resistant pathogens requires improved understanding of how resistance emerges and spreads in individual species, which are often embedded in complex microbial communities such as the human gut microbiome. Interactions with other microorganisms in such communities might suppress growth and resistance evolution of individual species (e.g., via resource competition) but could also potentially accelerate resistance evolution via horizontal transfer of resistance genes. It remains unclear how these different effects balance out, partly because it is difficult to observe them directly. Here, we used a gut microcosm approach to quantify the effect of three human gut microbiome communities on growth and resistance evolution of a focal strain of *Escherichia coli*. We found the resident microbial communities not only suppressed growth and colonisation by focal *E. coli* but also prevented it from evolving antibiotic resistance upon exposure to a beta-lactam antibiotic. With samples from all three human donors, our focal *E. coli* strain only evolved antibiotic resistance in the absence of the resident microbial community, even though we found resistance genes, including a highly effective resistance plasmid, in resident microbial communities. We identified physical constraints on plasmid transfer that can explain why our focal strain failed to acquire some of these beneficial resistance genes, and we found some chromosomal resistance mutations were only beneficial in the absence of the resident microbiota. This suggests, depending on in situ gene transfer dynamics, interactions with resident microbiota can inhibit antibiotic-resistance evolution of individual species.

## Introduction

The over- and inappropriate use of antibiotics has promoted the evolution of resistance in pathogens, resulting in a crisis for human healthcare [1]. To combat this problem, it is important to understand the underlying mechanisms of how resistance is acquired by bacteria and

for the whole-genome sequences of single colonies and PRJEB33429 for the 16S rRNA gene amplicon data. Other data are available through the Dryad repository: https://doi.org/10.5061/dryad. t1g1jwszq

**Funding:** Funding was received from the Swiss National Science Foundation project 31003A_165803 (http://www.snf.ch) (to MB and ARH). The funders had no role in study design, data collection and analysis, decision to publish, or preparation of the manuscript.

**Competing interests:** The authors have declared that no competing interests exist.

**Abbreviations:** IC90, concentration required to reduce growth by 90%; IS, insertion sequence; LB, lysogeny broth; MIC, minimal inhibitory concentration; PCoA, principal coordinate analysis; qPCR, quantitative PCR.

spreads within bacterial populations and communities [2,3]. A large body of research has used direct observations of resistance evolution in simplified laboratory conditions to understand how antibiotics drive the spread of resistance [4,5]. A key limitation of this approach is that it excludes interactions with other microorganisms, which we can expect to be important for bacteria evolving in natural or clinical settings because they spend most of their time in dense and diverse microbial communities. Interactions in species-rich microbial communities might negatively affect growth of individual species via, for example, competition for resources or niche space [6,7]. This may, in turn, inhibit antibiotic-resistance evolution of individual species, because reduced population growth should reduce the supply of new genetic variation. On the other hand, interspecific interactions also potentially have positive effects on growth and evolution of individual species via, for example, exchange of genetic material [8], cross-feeding, or public goods sharing [9–11]. Community-level interactions can also alter the strength of selection for resistant variants in the population [12,13]. In support of a key role for interspecific interactions in resistance evolution, observations of bacteria isolated from natural and clinical settings indicate genes involved in antibiotic-resistance evolution are often horizontally transferable [14–17]. Despite this, direct observations of how these different types of effects balance out are lacking. Consequently, it remains unclear how interactions with species-rich microbial communities affect growth and antibiotic-resistance evolution of individual species or strains of bacteria.

The impact of interactions with other microorganisms for antibiotic-resistance evolution is likely to be particularly important in the human gastrointestinal tract. This is one of the most densely inhabited environments in the world, colonised by a rich diversity of bacteria, viruses, and eukarya, which are embedded in a network of biotic interactions [18,19]. Interactions among microorganisms in the gut microbiome (which we take here to mean the resident microorganisms, their genes, and the local abiotic environment, following Marchesi and Ravel [20] and Foster and colleagues [19]) play an important role for human health [21]. For example, the microbiome minimises potential niche space for invading species, making it harder for them to establish in the community, thereby contributing to colonisation resistance against pathogens [22,23]. This suggests competitive interactions with other microorganisms are common, which we would expect to inhibit population growth of individual taxa and, in turn, constrain their ability to evolve antibiotic resistance. On the other hand, some interactions in the gut may be mutualistic [10] or modify the effects of antibiotics on individual species [24], potentially resulting in a net positive effect on growth. Moreover, recent metagenomic studies [25–27] showed the gut microbiome harbours a variety of mobile genetic elements, often carrying resistance and virulence genes, that are shared by community members. Consistent with this, horizontal transfer of resistance genes within individual hosts is central to resistance evolution in several key pathogens found in the gastrointestinal tract [16,28–30]. This suggests interactions with other microorganisms in the gut microbiome can also promote growth and resistance of individual taxa. We aimed to quantify the net effect of interactions with species-rich communities of other microorganisms, in particular those found in the human gastrointestinal tract, for growth and resistance evolution of a given strain that newly arrives in the community.

We approached this question using a human gut microcosm system consisting of anaerobic fermenters filled with human faecal slurry, including the resident microbial community and the beta-lactam antibiotic ampicillin, to which bacteria can evolve resistance by chromosomal mutations [31] or horizontal acquisition of beta-lactamase genes [32]. We used ampicillin because beta-lactam antibiotics are very widely used in human healthcare [33], resistance is a major problem [34], and key mechanisms by which bacteria evolve resistance to ampicillin overlap with resistance mechanisms against other antibiotics [35]. Because the microbiota in

faecal samples reflects the diversity of the distant human gastrointestinal tract [36], this approach allowed us to produce microcosms containing species-rich communities sampled from human gut microbiomes. We aimed to determine how interactions with this resident microbial community affected growth and resistance evolution of *E. coli*. We focused on *E. coli* because it is a ubiquitous gut commensal [37] and key opportunistic pathogen [38] for which antibiotic resistance is an increasing problem [39]. We inoculated each microcosm with a tagged, focal *E. coli* strain, before tracking its growth and resistance evolution in the presence and absence of ampicillin. By also including microcosms containing sterilised versions of the same faecal slurry (in which the resident microbial community had been deactivated), we quantified the net effect of interactions with the resident microbial community. This approach allowed us to (1) track growth and resistance evolution of the focal strain in the presence and absence of resident microbial communities sampled from several human donors; (2) isolate plasmid-carrying *E. coli* strains from the resident microbial community and identify constraints on horizontal transfer of resistance genes; and (3) characterise the resident microbial communities and how they changed over time. Our results show the resident microbial community inhibits both growth and resistance evolution of *E. coli*, despite the presence of resistance plasmids that can be conjugatively transferred to our focal strain in certain physical conditions.

## Results

### Resident microbial communities suppressed growth of a focal *E. coli* strain

We cultivated our focal *E. coli* strain in anaerobic microcosms in the presence and absence of an antibiotic and three different samples of gastrointestinal microbiomes, each from a different human donor, for 7 d (S1 Fig). On average, antibiotic treatment (ampicillin) decreased focal-strain abundance (effect of antibiotic in a generalised linear mixed model with zero inflation, glmmadmb, $\chi^2$ = 33.53, df = 1, $P$ < 0.001; Fig 1). For example, after 24 h, focal-strain abundance was reduced compared with ampicillin-free treatments by 69% (SD = 6.02) in the basal medium treatment, 78%–90% (depending on human donor) in the sterilised slurry treatments, and 84%–99.9% (depending on human donor) in the 'live' slurry treatments (Fig 1; S1 Table). Inclusion of the resident microbial community from human faecal samples also reduced focal-strain abundance on average, which we inferred by comparing the community treatments ('live' faecal slurries, including the resident microbial community) with the community-free treatments (sterilised versions of the same faecal slurries; effect of community in glmmadmb, $\chi^2$ = 6.65, df = 1, $P$ = 0.01; Fig 1).

The suppressive effect of resident microbial communities depended on both which human donor sample was used to prepare the microcosms (donor × community interaction in glmmadmb, $\chi^2$ = 10.23, df = 2, $P$ = 0.006) and the presence of ampicillin (antibiotic × community interaction in glmmadmb, $\chi^2$ = 5.2, df = 1, $P$ = 0.02), being strongest for populations exposed to both resident microbiota and the antibiotic (Fig 1). This resulted in extinction of the focal strain (below our detection limit) in populations exposed to ampicillin and the resident microbial communities from human donors 1 and 3. That is, in these treatments, the focal strain failed to colonise the community. For the resident community from human donor 2, the focal strain was driven to very low abundance in the presence of the community and ampicillin together but did not disappear completely (Fig 1). In the absence of ampicillin, resident communities still suppressed the focal strain on average (effect of community in glmmadmb for ampicillin-free treatments only, $\chi^2$ = 10.04, df = 1, $P$ = 0.002). As in the presence of ampicillin, the strength of this effect varied depending on human donor, being strongest for the resident microbial community from human donor 1 (effect of community for

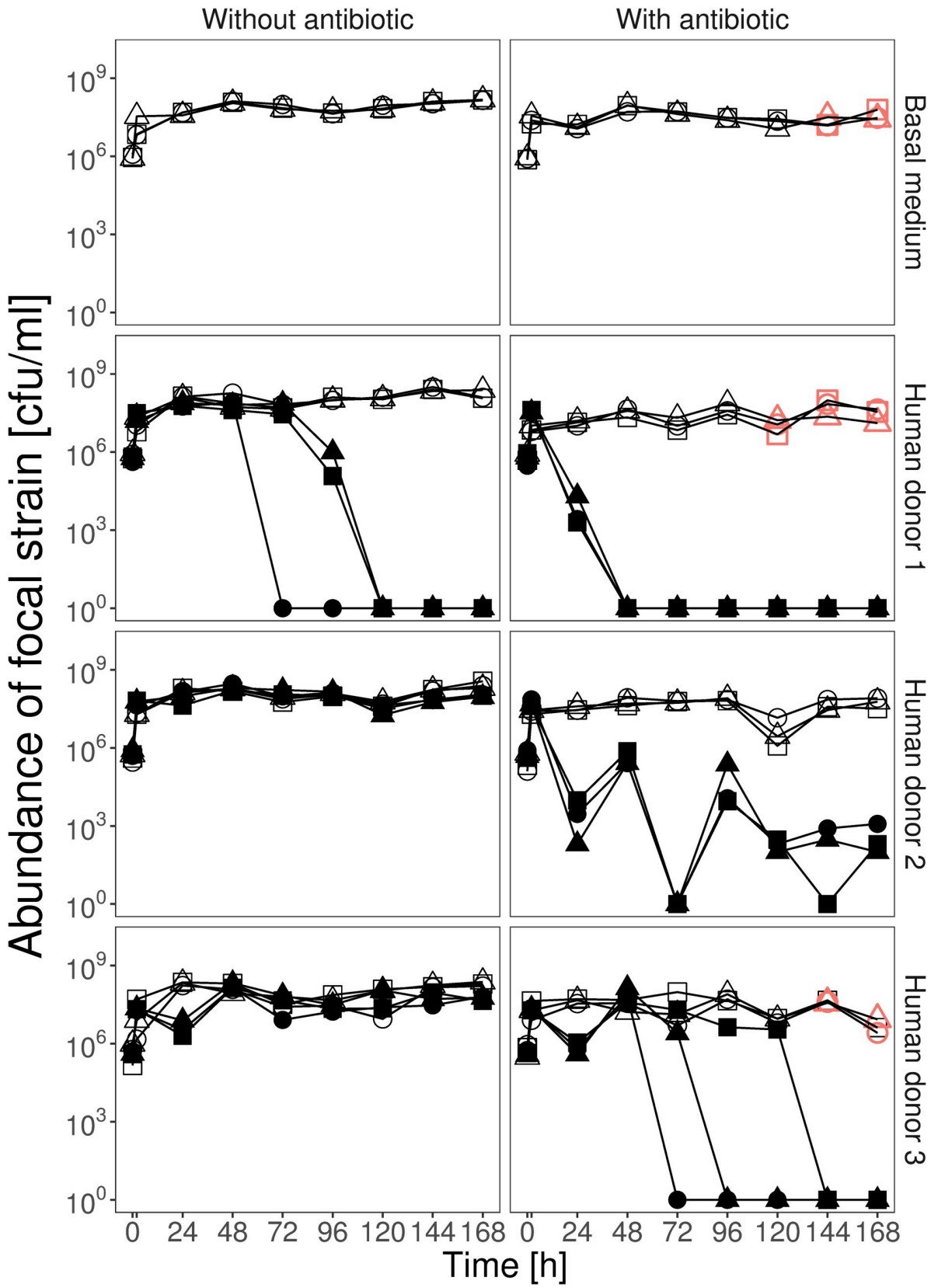

**Fig 1. Resident microbial communities suppressed growth and resistance evolution of focal *E. coli*.** Each panel shows abundance of the focal *E. coli* strain (in cfu per ml) over 7 d for either basal medium (top row) or with faecal slurry from one of three human donors, in the absence (left panels) or presence (right panels) of ampicillin, which was applied at the sampling time points after 2 h and thereafter at each daily transfer. Empty symbols show community-free treatments; filled symbols show treatments with the resident microbial community; red symbols show microcosms in which we detected ampicillin-resistant colony isolates of the focal strain. The three lines, each with different symbols, in each treatment show three replicate microcosms. Microcosms in which we detected no focal strain colonies are shown at $10^0$. Data are deposited in the Dryad repository: https://doi.org/10.5061/dryad.t1g1jwszq [40]. cfu, colony-forming units.

this donor in the absence of ampicillin: glmmadmb, $\chi^2 = 11.28$, df = 1, $P = 0.0002$; Fig 1 and S1 Table), resulting in exclusion of the focal strain. The resident community from human donor 3 suppressed average growth of the focal strain by approximately 54% across the entire experiment (effect of community for human donor 3 in the absence of ampicillin: glmmadmb, $\chi^2 = 4.77$, df = 1, $P = 0.03$; Fig 1 and S1 Table). For the resident community from human donor 2, average focal-strain abundance was lower in the presence of the community (mean reduction of 24% compared with community-free microcosms; Fig 1 and S1 Table), although this was not statistically significant (effect of community for human donor 2 in the absence of ampicillin: glmmadmb, $\chi^2 = 0.66$, df = 1, $P = 0.41$). We found no evidence that abiotic factors in the sterile faecal slurry were suppressive for the focal strain: there was no significant variation in average focal-strain abundance among the community-free treatments and the control treatment containing only the basal growth medium that was used to prepare faecal slurries (linear mixed-effects model, glmer: $\chi^2 = 0.41$, df = 3, $P = 0.94$). In summary, the resident microbial communities sampled from three human donors each suppressed growth of a focal *E. coli* strain in anaerobic fermenters filled with faecal slurry, but to varying degrees, and this effect was amplified by adding ampicillin.

## Stable total bacterial abundance but variable community composition over time

We used flow cytometry to estimate total bacterial abundance in microcosms containing resident microbial communities. In antibiotic-free microcosms, total bacterial abundance was approximately stable over time ($>10^9$ cells/ml; Fig 2) and was higher on average than in microcosms exposed to ampicillin (effect of antibiotic in a linear mixed-effects model, lmm: $\chi^2 = 10.37$, df = 1, $P = 0.001$). However, the suppressive effect of the antibiotic varied over time (antibiotic × time interaction in lmm: $\chi^2 = 101.81$, df = 7, $P < 0.001$), being strongest at the beginning of the experiment. The effect of the antibiotic also varied across communities from different human donors (antibiotic × donor interaction in lmm: $\chi^2 = 79.30$, df = 2, $P < 0.001$), with those from human donors 1 and 2 showing a stronger recovery after the first application of ampicillin (which resulted in a drop in abundance after 24 h) than the community from human donor 3. These results show our experimental setup sustained high numbers of microorganisms in the community treatments over time in both the presence and absence of ampicillin.

To investigate the community composition in these microcosms, we used amplicon sequencing of the variable regions 3 and 4 of the 16S rRNA gene. This revealed similar levels of within-sample diversity (Shannon's alpha diversity; Fig 3A) in microbiome samples from the three human donors at the start of the experiment. Within-sample diversity then decreased slightly over the first 24 h of the experiment and significantly between 24 h and 168 h (effect of time in a linear mixed-effects model including data from 24 h and 168 h, lme: $F_{1, 14} = 481.43$, $P < 0.001$). This applied across the three human donors (effect of human donor, lme: $F_{2, 15} = 2.07$, $P = 0.16$), which also showed similar shifts in taxonomic composition over time (Fig 3B). Communities at 0 h were dominated by the families Lachnospiraceae and Ruminococcaceae,

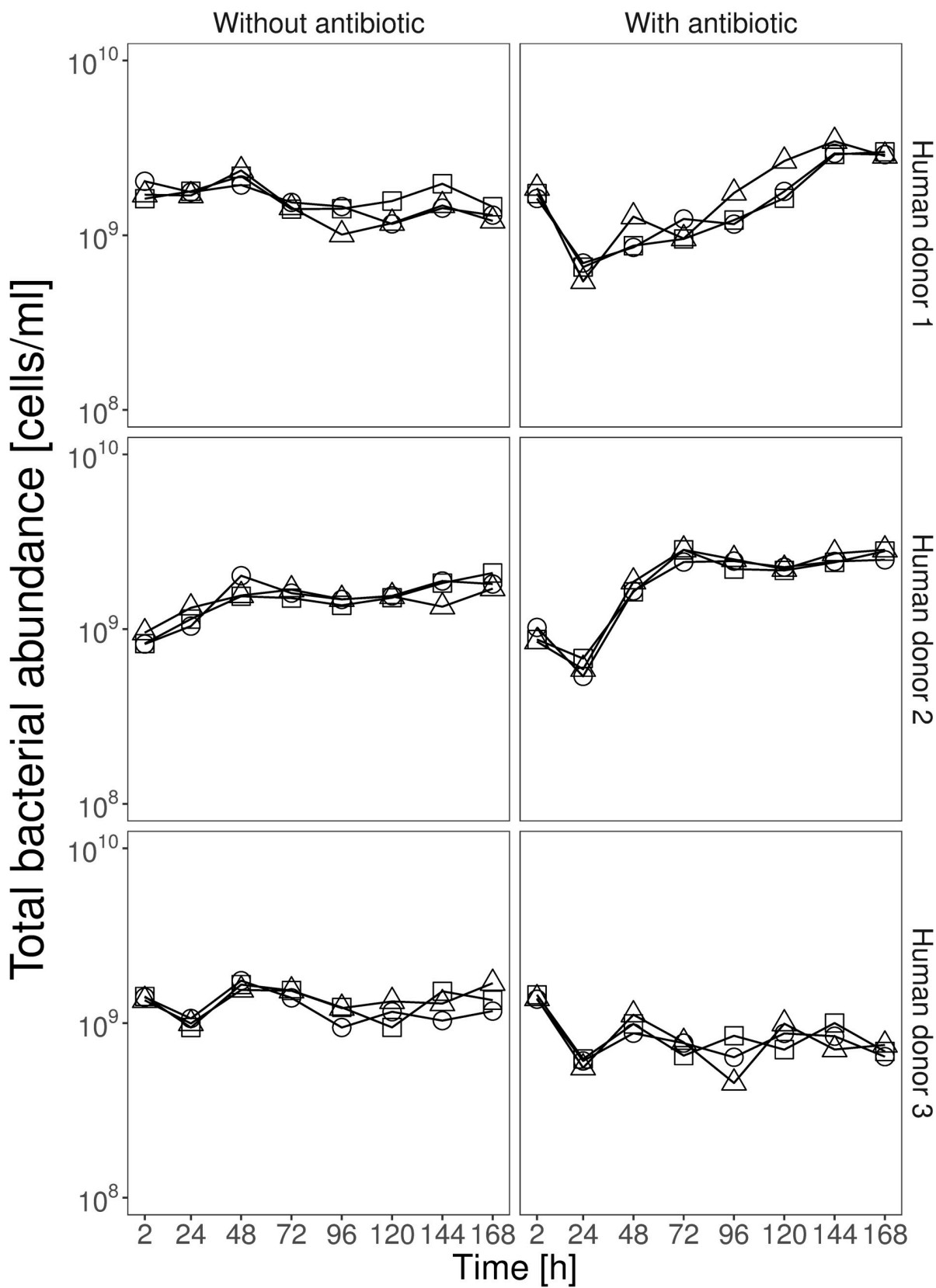

**Fig 2. Total bacterial abundance over time in community treatments with and without antibiotics.** Each row of panels shows data from one of the three human donors, and the right/left panels show treatments with/without ampicillin. The three replicate communities in each panel are shown by three different lines, each with a different symbol. Total bacterial abundance was measured by flow cytometry (see Material and methods). Data are deposited in the Dryad repository: https://doi.org/10.5061/dryad.t1g1jwszq [40].

plus Prevotellaceae for human donor 3. Over time, these groups became less abundant relative to Enterobacteriaceae and Bacteroidaceae. Analysis of the reads assigned to Enterobacteriaceae indicated *E. coli* was always the most abundant member of this group (see Material and methods), accounting for approximately 99% of the 16S rRNA amplicon reads assigned to Enterobacteriaceae in all samples (S2 Table), with the remaining approximately 1% comprising other species (e.g., *Enterobacter* sp., *Citrobacter* sp., and *Klebsiella* sp.). Compared with changes over time, ampicillin had a weak effect on within-sample diversity (effect of antibiotic: $F_{1,14}$ = 11.37, $P$ = 0.006). This interpretation was supported by an alternative analysis (principal coordinate analysis [PCoA]; S2 Fig) based on Bray-Curtis dissimilarities. Thus, despite approximately stable total abundance, we saw changes in community composition over time that were more pronounced than differences among communities from different human donors or antibiotic treatments. Despite these changes in relative abundance of different taxa, the identities of the top 5–6 families were stable over time and across human donors (Fig 3B).

We used quantitative PCR (qPCR) to better understand the contribution of resident *E. coli* and the focal strain to the total abundance of *E. coli* (see S1 Methods). Consistent with the amplicon sequencing data, this revealed increasing total abundance of *E. coli* sequences over time in both the presence and absence of ampicillin (S3 Fig). The copy number of focal-strain sequences relative to total *E. coli* indicated the focal strain was rare relative to other *E. coli* after 24 h (S3 Fig and S2 Table). At the end of the experiment, consistent with our estimates from selective plating and colony PCR, the focal strain was below the detection limit in treatments containing the community from human donor 1, both with and without ampicillin, and the human donor 3 community with ampicillin; in the other treatments, focal-strain sequences were rare compared with total *E. coli* (S2 Table).

## Antibiotic resistance evolved only in the absence of resident microbial communities

We screened for the emergence of antibiotic-resistant variants of the focal strain (that had acquired resistance to ampicillin) after every growth cycle by plating each population onto antibiotic-selective plates (8 μg/ml ampicillin; approximating the minimal inhibitory concentration [MIC] of the focal strain). We never observed resistant variants of the focal strain in any of the community treatments (populations exposed to the resident microbial communities from human microbiome samples). By contrast, in community-free treatments (basal growth medium and sterilised human faecal slurries), resistant variants appeared toward the end of the experiment at 120 h (slurry from human donor 1) and 144 h (basal growth medium and slurry from human donor 3), although not in sterilised samples from human donor 2 (Fig 1). Thus, the resident microbial community from human microbiome samples suppressed antibiotic-resistance evolution in our focal strain.

To investigate genetic mechanisms associated with resistance evolution and general adaptation to our experimental conditions, we performed whole-genome sequencing for two sets of focal-strain isolates from the final time point: eight ampicillin-resistant colony isolates from ampicillin plates (one from each of the eight populations in which we observed the emergence of antibiotic resistance during the experiment) and 33 randomly selected colony isolates from ampicillin-free plates (each from a different population and across all treatments). In the

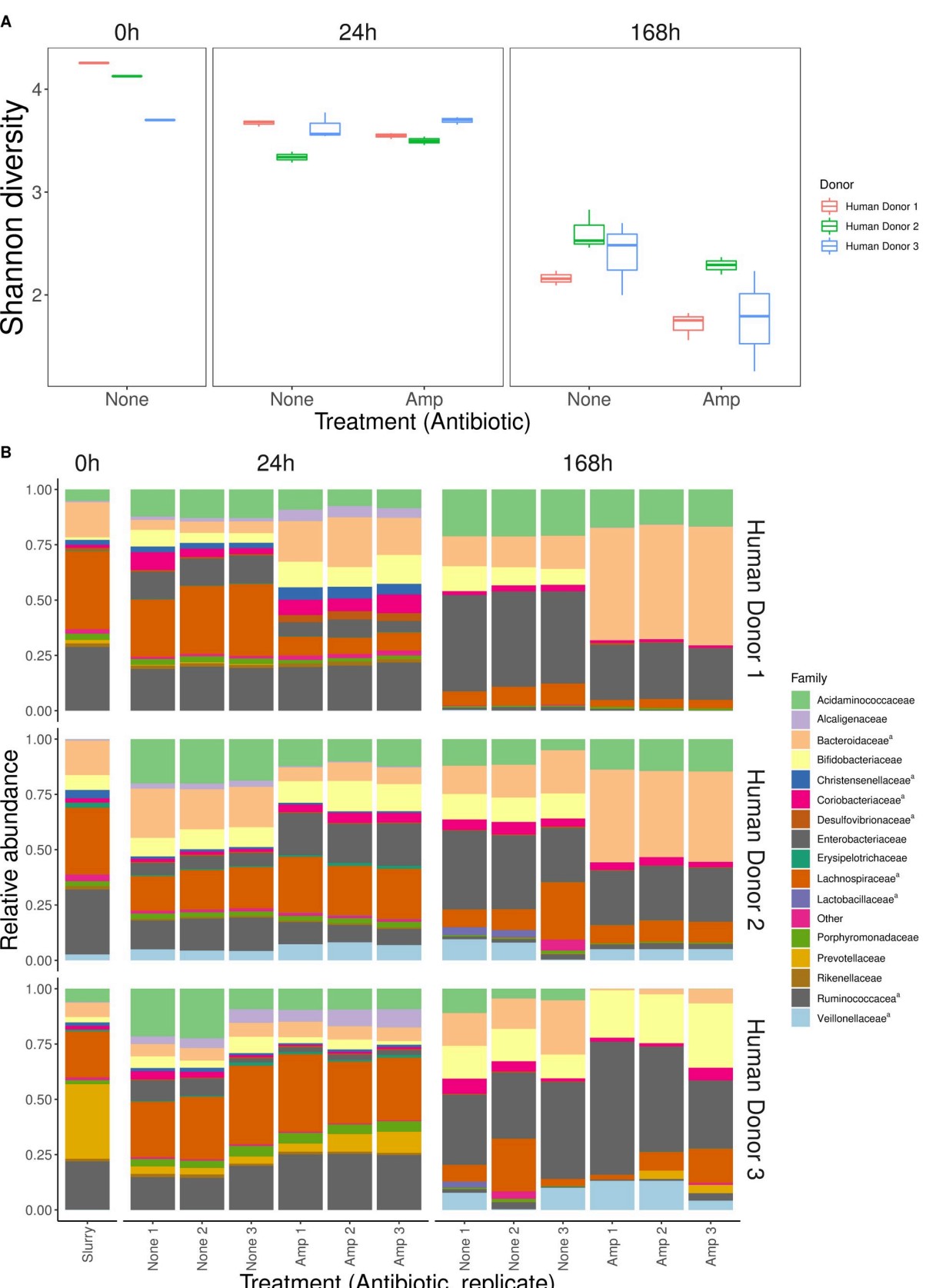

**Fig 3. Within-sample diversity and changes in community composition over time.** (A) Diversity is estimated here using Shannon's diversity index for samples from three time points (0 h, 24 h, and 168 h, shown at top) and three human donors (see legend), in the presence and absence of ampicillin ('Amp', x-axis). Each box shows data from three replicate microcosms per treatment group and time point (except for at 0 h, which shows the single initial sample from each human donor). (B) Relative abundance of the 15 most prevalent bacterial families in each microcosm at three time points (0 h, 24 h, and 168 h, shown at top) for each human donor (rows of panels, labelled at right) in the presence and absence of ampicillin (three replicates each treatment; x-axis). Superscript 'a' in the legend indicates obligately anaerobic families. Data are deposited in the European Nucleotide Archive under the study accession number PRJEB33429.

antibiotic-resistant isolates, all SNPs and the deletion we found (Table 1) were in genes related to membranes (*ompR*, *ftsI*, *opgB*), stress responses (*relA*), or transcription (*rpoC*, *rpoD*). Two genes were mutated independently in multiple colony isolates: *rpoC* and *rpoD*. We also detected an insertion sequence (IS) movement between *perR* and *insN* in two colony isolates. Of genes in which we detected mutations in a single colony isolate, *ftsI* [41], *relA* [42], and *ompR* [43] have each been previously annotated as being involved in resistance to beta-lactam antibiotics. *ompR* was also mutated in three randomly selected colony isolates, all from populations that had been exposed to ampicillin during the experiment (S3 Table). We found six other genes mutated in parallel in between two and five randomly selected colony isolates (S3 Table). Five of these were mutated in isolates from both antibiotic and antibiotic-free treatments. This included *insN* and *gtrS*, both mutated in five isolates. Across the two sets of colony isolates (randomly selected and antibiotic resistant), three other loci were mutated in both sets. These were *rpoD* (only in isolates that had been exposed to antibiotics), *opgB*, and *yaiO* (both in isolates from antibiotic and antibiotic-free treatments). In summary, we found some parallel genetic changes specific to antibiotic treatments and consistent with known resistance mechanisms, plus other genetic changes that occurred across antibiotic and antibiotic-free treatments and are therefore more likely involved in general adaptation.

## Plasmid acquisition was constrained by lack of transfer, not lack of fitness benefits

We next sought to explain why we never observed antibiotic-resistance evolution of the focal strain in the presence of resident microbial communities, which we had expected to harbour beneficial resistance genes [26,27,44–46]. We hypothesised this could have been due to a lack of horizontally transferable resistance genes in the resident microbial communities. However, we detected ampicillin-resistant *E. coli* in the resident microbial communities from human donors 1 and 3 (by selective plating), and after sequencing their genomes, we identified several antibiotic-resistance genes that were associated with plasmid genes (Fig 4 and S4 Table). In the hybrid assembly (using MinION and Illumina reads) of a representative isolate from human donor 1, we identified two plasmids. The larger plasmid had four known resistance genes (Fig 4A), including one conferring resistance to beta-lactam antibiotics. We also identified three IncF replicons on this plasmid and a complete set of *tra* genes, which are involved in conjugative plasmid transfer. The second plasmid carried a known replicon (ColRNAI) and mobilisation genes (*mbeA* and *mbeC*), plus a complete colicin E1 operon (*cnl*, *imm*, *cea*). For a representative isolate from human donor 3, we found the plasmid replicon and resistance genes integrated on the chromosome (Fig 4B). This putative integrated plasmid from human donor 3 also carried multiple resistance genes, including a beta-lactamase and an IncQ replicon, which is a part of the *repA* gene [47], but we detected no *tra* genes. The other five genome assemblies (Illumina reads for other isolates from the same human donor) contained the same resistance genes and replicons across multiple smaller contigs (S4 Table). Mapping the corresponding sequencing reads of all Illumina-sequenced isolates to the long-read data single contig found in the isolates sequenced on the MinION platform revealed identical mapping in all

**Table 1. Genes mutated in ampicillin-resistant colony isolates of the focal strain from the end of the experiment.** Data are deposited in the European Nucleotide Archive under the study accession number PRJEB36309.

| Treatment group | Human donor | Replicate | afuB | afuC | cspB<>cspF | cyoA | fecI | ftsI | insA<>uspC | insB1 | insD1 | ompR | opgB* | perR | perR<>insN | relA | rpoC | rpoD | yaiO* | yehE | yjhD<>insO | yjhD<>yjhE |
|---|---|---|---|---|---|---|---|---|---|---|---|---|---|---|---|---|---|---|---|---|---|---|
| Basal medium with antibiotic | None | 1 | | | x | | | • | | | | | | | x | | | | | | | x |
| | | 2 | | x | | | | | | | | | | | | | | | | x | | |
| | | 3 | | | | | | | | | | | | | x | | | | | | | |
| Community-free with antibiotic | 1 | 1 | | | | | | | x | | x | | | x | | • | • | • | | | | |
| | | 2 | | | | | | | | | | • | | | | | | | | | | |
| | | 3 | x | | | x | | | | | | | • | | | | | | | | | |
| Community-free with antibiotic | 3 | 1 | | | | | | | | x | | | | | | | | • | x | | | |
| | | 2 | | | | | x | | | | | | | | | | ▲ | | | | x | |

*Genes also mutated in randomly selected clones from ampicillin-free plates (see S3 Table).

▲ Deletion.

x Insertion.

• SNP.

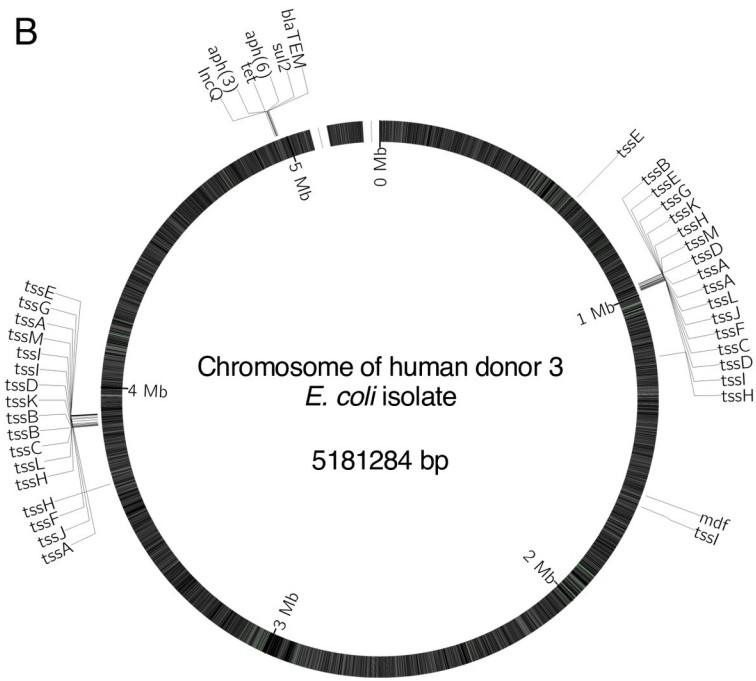

**Fig 4. Schematic maps of plasmids and chromosomes for representative resident *E. coli* isolates from (A) human donor 1 and (B) human donor 3.** Sequences are annotated with known plasmid replicon sequences (IncFIA, IncFIB, IncFIC on plasmid 1 and ColRNAI on plasmid 2 in [A]; IncQ on the chromosome in [B]), genes involved in horizontal transfer (*tra* and *trb*), known resistance genes (*bla*$_{TEM}$, *sul2*, *aph(3)*, *aph(6)*, *mdf*), genes involved in type VI secretion systems (*tss*, *vgrG*), and genes involved in colicin production and immunity (*cea*, *cnl*, *imm*) and mobilisation (*mbeA* and *mbeC*). The genome of the isolate from human donor 3 (B) is not closed, as indicated with a gap. Colours indicate coding (black) and noncoding regions (green); note the scale varies among chromosomes and plasmids.

10 cases, consistent with these genomes having the same structure across each of the isolates (S4 Table).

We hypothesised that the lack of plasmid-driven resistance evolution in our focal strain might have been caused by constraints on conjugative transfer that made these plasmids inaccessible. Using a conjugative mating assay on agar, we never found transconjugants of our focal strain when it was mixed with an isolate from human donor 3 (identified above as carrying a putative integrated plasmid). This is consistent with the lack of *tra* genes on this plasmid and suggests it could not be transferred into our focal strain by conjugation in the absence of other drivers of horizontal gene transfer (e.g., phages or other plasmids). This is also consistent with past work suggesting IncQ plasmids are mobilisable rather than conjugative [48,49] and that we did not detect any other plasmid replicons in the same isolates. However, for the plasmid from human donor 1, we found transconjugants of our focal strain at the end of the mating assay, which we confirmed by colony PCR (S4 Fig). This suggests this plasmid was conjugative and could be transferred to our focal strain, consistent with the presence of *tra* genes on this plasmid (Fig 4A).

Given that the resistance plasmid from human donor 1 was transferable into our focal strain, why did it not spread in the main experiment above? We hypothesised this could result from plasmid-borne resistance being less beneficial than resistance acquired by chromosomal mutation (as we observed in community-free treatments in the main experiment). We would expect a net benefit of resistance to result in increased population growth at the antibiotic concentration applied during the experiment (7.2 μg/ml). We found acquisition of the plasmid conferred a much larger increase in population growth across all nonzero antibiotic concentrations than that observed for evolved colony isolates that had acquired resistance via chromosomal mutation during the main experiment (Fig 5A and S5 Table). Furthermore, in pairwise competition experiments, the transconjugant carrying this plasmid had a strong competitive advantage relative to the wild type in the presence of the resident microbial community from human donor 1 (S5A Fig). This fitness advantage was increased by adding ampicillin at the concentration we used in the main experiment and even further by adding ampicillin at three times the IC90 (concentration required to reduce growth by 90%) of the ancestral focal strain (community × ampicillin interaction by permutation test; *P* = 0.029; S5B Fig). This shows it would have been highly beneficial for the focal strain to acquire the plasmid in our experiment, particularly in the presence of ampicillin.

Unlike transconjugants carrying the plasmid from resident *E. coli*, two evolved colony isolates of the focal strain carrying chromosomal resistance mutations had a fitness advantage relative to the wild type only in the absence of the community; in the presence of the resident microbial community, they had a fitness disadvantage (effect of community by permutation test: *P* < 0.001 for isolates from human donors 1 and 3; S5B Fig). This suggests the genetic changes associated with increased resistance in these isolates in the absence of resident microbiota would not have been beneficial in the Community + Ampicillin treatments of the main experiment, unlike the plasmid we isolated from resident *E. coli*. This conclusion was supported by comparing transconjugants carrying the resistance plasmid and evolved colony isolates from human donors 1 and 3 (S6 Fig). Further competition experiments showed the

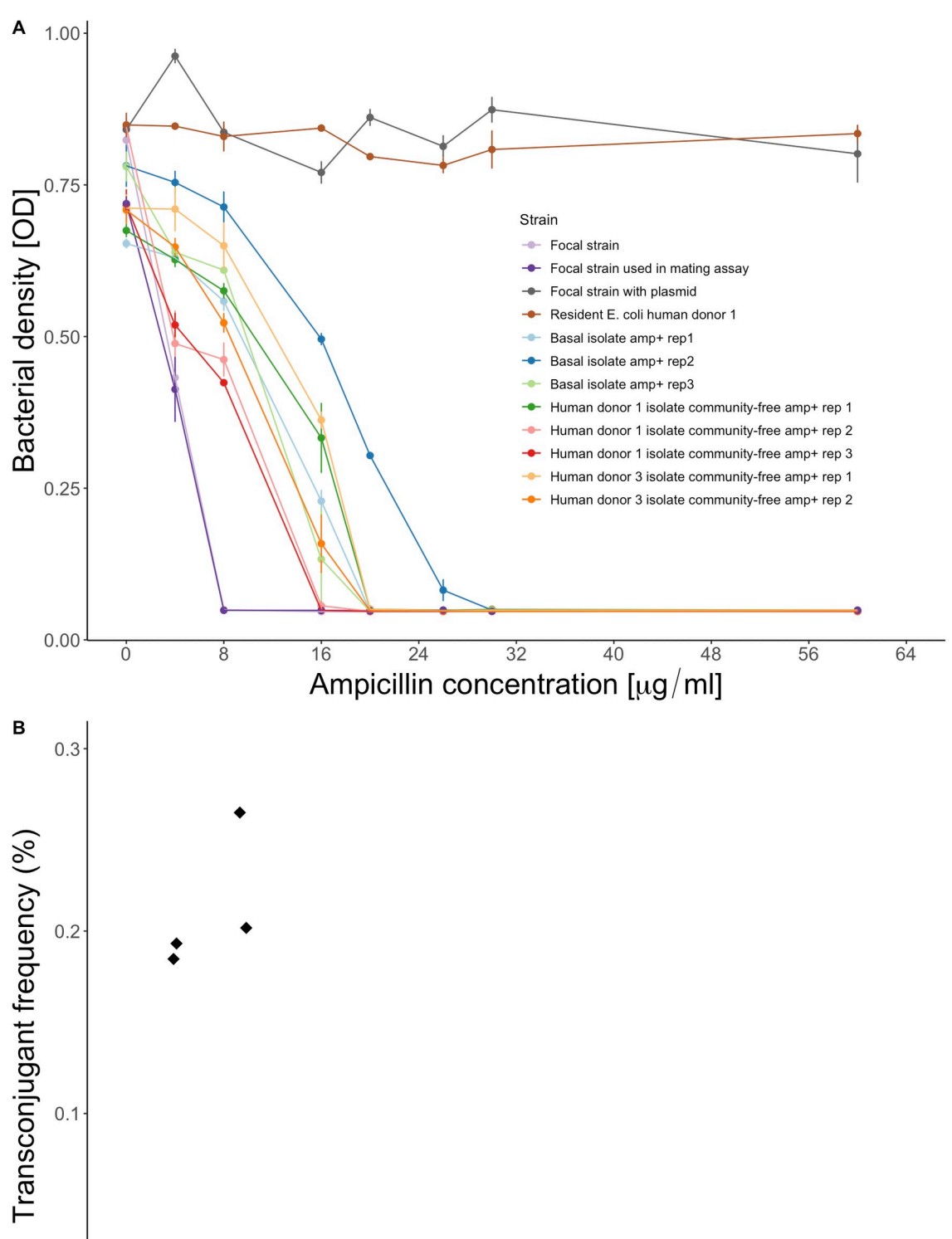

**Fig 5. Transfer of a resistance plasmid from the resident microbial community is sensitive to abiotic conditions, and it confers a large increase in resistance.** (A) Antibiotic susceptibility of the ancestral focal strain, the version of the focal strain used to isolate transconjugants (see Material and methods), the focal strain with the plasmid (transconjugant), the resident *E. coli* isolate used as the plasmid donor, and eight evolved focal-strain colony isolates that we isolated on ampicillin ('amp') plates and that had chromosomal mutations (S2 Table). Average OD values ± SE are shown after 24-h growth at each ampicillin concentration. (B) Transconjugant frequency (as a percentage of the total recipient population) after mating experiments in various conditions. The recipient strain was a tagged version of the ancestral focal strain and the plasmid donor was a resident *E. coli* isolate from human donor 1. For the faecal slurry treatment, we used sterilised faecal slurry from human donor 1. Data are deposited in the Dryad repository: https://doi.org/10.5061/dryad.t1g1jwszq [40]. LB, lysogeny broth; OD, optical density; rep, replicate population.

competitive advantage of resident *E. coli* from human donors 1 and 3 carrying resistance genes was also present in the absence of other resident microbiota (S7 Fig).

Another possible explanation for the lack of transfer of the resistance plasmid from human donor 1 in the main experiment is that conjugative transfer might be specific to particular environmental conditions. This has been observed for other plasmids across various experimental conditions [50,51]. We tested this by mating assays as above, but in a range of different experimental conditions. We detected transconjugants that had acquired the plasmid at a final frequency of approximately 0.2% (as a fraction of the total recipient population) after mixing the plasmid-carrying isolate from human donor 1 and the focal strain on an agar surface, but we found no transconjugants after doing the same experiment in three different types of liquid growth medium (lysogeny broth [LB], anaerobic LB, and anaerobic community-free faecal slurry; Fig 5B). This shows transfer of the conjugative plasmid we isolated from human donor 1 requires particular abiotic conditions, which may explain why our focal strain failed to evolve resistance via horizontal gene transfer in the presence of resident microbial communities. This was supported by simulations of a hypothetical plasmid with similar properties but that is transferable in our gut microcosm system (S1 Model). This indicated that, if the plasmid from human donor 1 had been conjugatively transferable in our gut microcosm system, we would have detected transconjugants in our main experiment (although only with relatively high transfer rates; S1 Model). The same model also showed growth suppression of invading lineages by resident microbiota can reduce transconjugant abundance, suggesting even when horizontal acquisition of beneficial resistance genes is common, interactions with resident microbiota can impede their spread.

## Discussion

We found resistance via chromosomal mutation to an important class of antibiotics (beta-lactams) evolved in a focal *E. coli* strain in our experiment only in the absence of resident microbial communities sampled from healthy human volunteers. The suppressive effect of these resident microbial communities was strong enough that the focal *E. coli* strain was driven towards extinction (below our detection limit) when it was exposed to both ampicillin and the community simultaneously (with communities from two of the three human donors we tested). Consequently, the net effect of the resident microbial communities here was to confer a form of colonisation resistance against a nonresident strain and to prevent that strain from evolving antibiotic resistance. Our analysis of resident *E. coli* isolates (not the focal strain) from the microbial communities showed this occurred despite the presence of beneficial, potentially horizontally transferable resistance genes. Genomic analyses and conjugation experiments with these resident *E. coli* isolates showed the in situ transfer dynamics depend critically on genetic (the presence of genes encoding the machinery for conjugative transfer) and abiotic (physical structure of the environment) factors. This is important because ultimately, it is the in situ transfer dynamics that will determine whether or not horizontal transfer of beneficial resistance genes is sufficient to counteract the growth-suppressive effects of

interactions with the community and confer a net benefit to invading lineages. Community-level interactions also modified selection for resistance, amplifying growth inhibition by ampicillin and altering the relative advantages/disadvantages of individual resistant genotypes (evolved isolates with chromosomal mutations conferring relatively weak increases in ampicillin resistance had a reduced advantage, but plasmids from resident *E. coli* that conferred relatively large increases in ampicillin resistance were more beneficial in the presence of resident microbiota). Overall, this indicates resident microbiota influence resistance evolution of invading strains via effects on both population dynamics and the strength of selection for resistance.

The first key implication of our work is that as well as suppressing growth and colonisation by invading strains [23,52], the gastrointestinal microbiota can inhibit antibiotic-resistance evolution. There are several possible ecological mechanisms by which the microbiota may suppress growth of invading lineages [53], such as niche and nutrient competition [54], direct killing via bacteriocins [22,55], phage production [56], or changing the concentration of compounds such as primary and secondary bile acids [57,58]. It was not our aim to pull apart the mechanisms by which resident microbiota suppress invading bacteria (studied in more detail elsewhere; [55,59]). Nevertheless, our data on community structure indicate resident Enterobacteriaceae, including *E. coli*, had a competitive advantage in our system, potentially explaining suppression of the focal strain. This was further evidenced by the advantage of transconjugants carrying plasmids from resident *E. coli* (S5 Fig) and resident *E. coli* over our ancestral focal strain in competition experiments (S7 Fig). The competitive advantage of resident *E. coli* extended to ampicillin-free conditions in pure culture. Possible contributors to this include type VI secretion systems we detected in the genomes of the human donor 1 and 3 *E. coli* isolates and a colicin plasmid present in human donor 1 *E. coli* isolates (Fig 4). In supernatant experiments, we did not find evidence of direct inhibition via phages in the community samples (see S1 Methods). Crucially, no matter how interactions with the microbiota suppress growth of an invading lineage, we expect the reduced population size, growth, and replication to, in turn, reduce the supply of new genetic variation. This is consistent with in vitro work with malaria parasites showing competition between two species under resource limitation impeded drug-resistance evolution [60] and previous studies with *Pseudomonas fluorescens* showing that a eukaryotic predator [61] or a bacteriophage [62] can suppress the emergence of antibiotic resistance. Thus, suppression of an invading lineage via interactions with resident microbiota may frequently have a knock-on effect on resistance evolution.

A second key implication is that resident microbiota modified selection on antibiotic resistance in our focal strain. The stronger effect of ampicillin on focal-strain growth in the presence of resident microbiota (Fig 1 and S1 Table) indicates resistance would have been more beneficial here. In support, the benefits of resistance plasmid acquisition were greatest in the presence of resident microbiota (S5 Fig). This is counter to the expectation that antibiotics may be less effective in more dense communities because of an 'inoculum effect' [63]. We found inhibition of our focal strain was indeed altered by very high *E. coli* abundance in pure cultures (S8 Fig), although there was still considerable inhibition even at the highest densities. This indicates such inoculum effects were weaker in the presence of species-rich communities than in pure cultures of *E. coli* and/or were counterbalanced by opposing effects of community-level interactions on ampicillin inhibition of *E. coli*. By contrast, chromosomal resistance mutations that emerged in the absence of resident microbiota (and which conferred relatively small increases in resistance in pure culture) were no longer beneficial in the presence of resident microbiota, indicating a larger change in resistance is needed to overcome the relatively strong effect of ampicillin here. This complements recent work showing natural communities from pig faeces can increase costs of antibiotic resistance for individual species [12] and that

costs of phage resistance can be altered by interactions with other bacterial species [64]. More generally, this supports the notion that community-level interactions modulate the costs and benefits of antibiotic resistance via mechanisms that are only just beginning to be understood [65].

A third key implication of our data concerns the genetic and environmental constraints on horizontal gene transfer that determine whether or not the growth-suppressive effects of the microbiota are counterbalanced by horizontal transfer of beneficial genes. The unavailability of known resistance genes to the invading focal strain in the community from human donor 3 was because they were integrated in the chromosome. The plasmid we isolated from the human donor 1 community was conjugative, but transfer depended on the abiotic conditions. This suggests the potential for plasmid transfer to allow invading lineages to overcome the suppressive effects of the microbiota depends critically on whether they are conjugative (which can be predicted from sequence data) and on the sensitivity of conjugative transfer to local physical conditions (which is harder to predict from sequence data). Consistent with this, previous research has shown conjugative transfer in *E. coli* and other species is sensitive to the physical experimental conditions [51,66]. Furthermore, mating pair formation machinery, usually encoded by the plasmid, in some cases promotes biofilm formation, which can, in turn, promote the spread of plasmids [67]. This raises the question of whether some plasmids have evolved to manipulate the physical structure of bacterial populations to promote transfer. Despite these constraints, plasmids are clearly sometimes transferred in vivo, as has been observed in animal models [68,69] and human gut microbiomes [32,44,45]. In line with plasmids being key vectors of beta-lactamases [70], the conjugative plasmid we identified was highly effective in terms of resistance. This suggests plasmid-borne resistance will be under strong positive selection once established and can spread rapidly via clonal expansion. However, our experiment showed the initial horizontal transfer required for such spread is sensitive to genetic and abiotic constraints.

Our approach allowed us to isolate the effect of interactions between diverse microbial communities and a focal *E. coli* strain. Our amplicon sequence data showed the communities had a representative taxonomic composition for healthy human donors. There were still diverse communities present at the end of the 7-d experiment, albeit with a change in the relative abundance of different taxa. The observed rise of Enterobacteriaceae and Bacteroidaceae has been seen in other experiments with gastrointestinal communities (e.g., [71]) and might be explained by the nutrient content of the medium, micromolar oxygen levels, or such in vitro systems favouring faster population growth [72]. In conditions where antibiotics are applied at higher concentrations or affect a greater fraction of extant taxa, we can expect stronger shifts in community composition in response to antibiotic treatment. We used a sublethal concentration in our main experiment to allow us to track invasion and growth by the focal strain. Nevertheless, in competition experiments at higher antibiotic concentrations, we saw similar outcomes in that plasmids from the resident microbiota were highly beneficial, whereas chromosomal mutations were not (S5 Fig). More importantly, the shift in community composition over the 7-d experiment does not explain the observed suppression of the focal strain, because this was already visible after 1 d.

Although our experimental system likely differs from the gastrointestinal tracts these bacteria were isolated from in ways that affect community composition, cultivating them in vitro allowed us to quantify the effect of species-rich communities sampled from gastrointestinal tracts on resistance evolution of a relevant opportunistic human pathogen. A key limitation of our study is the sample size (three human donors, one focal strain, one antibiotic). Some outcomes might change with different types of resident microbiota or different types of plasmids (explored in the S1 Model). Nevertheless, we observed a qualitatively consistent suppression of

the focal strain across the three human donors, which was always stronger in the presence of ampicillin and, in some cases, was associated with colonisation resistance (extinction of the focal strain). Additionally, we chose *E. coli* and ampicillin because they are both important for understanding resistance evolution in nature and share some important properties in this respect with other bacteria and antibiotics (our rationale is explained further in the Introduction). Despite the low sample size, we observed a qualitatively consistent suppression of the focal strain across the three human donors, which was always stronger in the presence of ampicillin and in some cases was associated with colonisation resistance (extinction of the focal strain). A key challenge for future work will be to uncover the aspects of microbiome composition (e.g., presence/absence of particular taxa) that determine colonisation resistance against invading species and influence antibiotic resistance, whether these are specific to particular invading species/antibiotics, and how such interactions are modified in vivo by local spatial structure [73] and immune responses [74]. Indeed, interactions mediated via the host immune system are another possible mechanism of colonisation resistance [75–77].

In conclusion, we showed species-rich microbial communities sampled from human gastrointestinal tracts can suppress growth and resistance evolution of an invading lineage. Given the variety and likely common occurrence of mechanisms that can generate such suppression of invaders (e.g., resource competition), these types of effects are probably common in species-rich communities such as the mammalian gastrointestinal tract. Crucially, resident microbiota also altered the strength of selection for resistance (ampicillin was more suppressive for the focal strain in community treatments) and the fitness effects of individual genetic changes (high-level resistance plasmids became more beneficial in the community treatments, but low-level resistance mutations became less beneficial). Our other data and simulations showed that whether the growth-suppressive effects of resident microbiota are counterbalanced by beneficial horizontal gene transfer depends on genetic and environmental constraints that can impede the spread of resistance plasmids. This has important implications for the prediction of resistance evolution from genetic and metagenomic data, such as those widely collected through surveillance efforts [78,79]: identifying mobile resistance genes in a diverse community is not enough to predict resistance evolution, requiring in addition information about genetic and environmental constraints on in situ transfer dynamics.

## Material and methods

### Ethics statement

The stool samples used in this study were from anonymous, consenting human donors and the sampling protocol was approved by the ETHZ Ethics Commission (EK 2016-N-55).

### Bacterial strains

We used *E. coli* K12 MG1655 carrying a streptomycin-resistance mutation (*rpsL* K43R) as the focal strain. Two days prior to the experiment, we streaked the focal strain on LB agar (Sigma-Aldrich, Buchs, Switzerland) and incubated overnight at 37˚C. To incubate the focal-strain cultures anaerobically prior to the microcosm experiment, we prepared 42 Hungate tubes (VWR, Schlieren, Switzerland) with LB (Sigma-Aldrich), which was supplemented with 0.5 g/l L-Cysteine and 0.001 g/l Resazurin (reducing agent and anaerobic indicator, respectively), flushed the headspace with nitrogen, sealed the tubes with a rubber stopper, and autoclaved them. One day before the experiment, we randomly picked 42 colonies and inoculated them in the 42 Hungate tubes containing anaerobic LB and incubated at 37˚C overnight with 220-rpm shaking. We then used these 42 independent cultures of the focal strain to inoculate the main experiment described below.

## Human microbiome samples

All stool samples were collected at the Department of Environmental Systems Science, ETH Zürich, on 15 May 2018. Inclusion criteria were older than 18 y, not obese, not recovering from surgery, and no antibiotics in the last 6 mo. Each sample was collected in a 500-ml plastic specimen container (Sigma-Aldrich) and kept anaerobic using one AnaeroGen anaerobic sachet (Thermo Scientific, Basel, Switzerland). The three samples used for the experiment were randomly selected from a larger number of donated samples. We collected the samples in the morning before the experiment and kept them for maximum 1 h before processing. To prepare faecal slurry from each sample, we resuspended 20 g of sample in 200 ml anaerobic peptone wash (1 g/l peptone, 0.5 g/l L-Cysteine, 0.5 g/l bile salts, and 0.001 g/l Resazurin; Sigma-Aldrich) to prepare a 10% (w/v) faecal slurry. We then stirred the slurry for 15 min on a magnetic stirrer to homogenise, followed by 10 min of resting to sediment. At this point we removed 100 ml of each faecal slurry ('fresh slurry'), which we used later to reintroduce the resident microbial community to sterilised slurry (for the community treatments). To sterilise the faecal slurries, we transferred 100 ml to a 250-ml Schott bottle, flushed the headspace with nitrogen gas, sealed them with rubber stoppers, and autoclaved for 20 min at 121˚C.

## Inoculating anaerobic gut microcosms, sampling, and bacterial enumeration

For the start of the experiment (S1 Fig), we filled 42 Hungate tubes with 7 ml of basal medium, which was based on earlier studies [80,81] with some modifications (2 g/l Peptone, 2 g/l Tryptone, 2 g/l Yeast extract, 0.1 g/l NaCl, 0.04g $K_2HPO_4$, 0.04 g/l $KH_2PO_4$, 0.01 g/l $MgSO_4x7H_2O$, 0.01 g/l $CaCl_2x6H_2O$, 2g/l $NaHCO_3$, 2 ml Tween 80, 0.005 g/l Hemin, 0.5 g/l L-Cysteine, 0.5 g/l bile salts, 2g/l Starch, 1.5 g/l casein, 0.001g/l Resazurin, pH adjusted to 7, addition of 0.001g/l Menadion after autoclaving; Sigma-Aldrich), and for the subsequent re-inoculation cycles with 6.5 ml of basal medium. We flushed the headspace of each tube with nitrogen gas, sealed it with a rubber septum, and autoclaved to produce anaerobic microcosms containing only basal medium.

On day 1 of the experiment, we introduced faecal slurry and antibiotics to each tube according to a fully factorial design (S1 Fig), with three replicate microcosms in each combination of Human Donor (1, 2 or 3), Community (present or absent), and Antibiotic (with or without). In the community-free treatments, we added 850 μl of sterile slurry. In the community treatments, we added 350 μl of fresh slurry and 500 μl of sterilised slurry. In the antibiotic treatment, we added ampicillin to a final concentration of 7.2 μg/ml, approximating the IC90 for the focal strain; this was introduced 2 h after the focal strain had been inoculated (8 μl of focal *E. coli* from one of the 42 overnight cultures introduced at 0 h; approximately 1:1,000 dilution). As a control treatment testing for the effect of sterilised slurry, we also inoculated the focal strain into three replicate microcosms containing only the basal medium (supplemented with 850 μl of peptone wash to equalise the volume with the slurry treatments), and we did this with and without antibiotic treatment. We incubated all microcosms at 37˚C in a static incubator. After 24 h, we transferred a sample of 800 μl from each microcosm to a new microcosm (containing basal medium plus 500 μl of sterile slurry from the corresponding human donor in the community and community-free treatments, or basal medium plus peptone wash for the basal medium treatment, supplemented with ampicillin at each transfer in the antibiotic treatments), and we repeated this for 7 d.

To estimate the abundance of the focal strain during the experiment, we used a combination of selective plating and colony PCR. For selective plating, we serially diluted the samples and plated them on Chromatic MH agar (Liofilchem, Roseto degli Abruzzi, Italy), which

allowed us in a first step to discriminate *E. coli* from other species based on colony colour. By supplementing these agar plates with streptomycin (200 µg/ml), to which our focal strain is resistant, we selected against other *E. coli* that were not resistant to streptomycin. To screen for variants of our focal strain that acquired resistance to ampicillin during the experiment, we additionally plated each sample onto the same agar supplemented with both streptomycin (200 µg/ml, Sigma-Aldrich) and ampicillin (8 µg/ml, Sigma-Aldrich). We did this after every growth cycle. Despite initial screening of microbiome samples revealing no resident *E. coli* that could grow on our selective plates, later in the experiment we found such bacteria to be present in some samples (that is, non-focal-strain *E. coli* that could grow on our plates and were presumably very rare at the beginning of the experiment). To discriminate between these *E. coli* and our focal strain, we used colony PCR with specific primers (forward [5′-AGA CGA CCA ATA GCC GCT TT-3′]; reverse [5′-TTG ATG TTC CGC TGA CGT CT-3′]). For colony PCR, we picked 10 colonies randomly for each time point and treatment. The PCR reaction mix consisted of 2x GoTaq green master mix, 2.5 µM of each primer, and nuclease free water. The thermal cycle programme ran on a labcycler (Sensoquest, Göttingen, Germany) with 6-min 95˚C initial denaturation and the 30 cycles of 95˚C for 1 min, 58˚C for 30 s, 72˚C for 35 s, and a final elongation step of 72˚C for 5 min. For gel electrophoresis, we transferred 5 µl of the PCR reaction to a 1.5% agarose gel stained with SYBR Safe (Invitrogen, Thermo F. Scientific) and visualised by UV illumination. Focal-strain abundance was then estimated by multiplying the frequency of the focal strain determined by colony PCR with the total colony count for each plate. To account for the possibility that the focal strain was still abundant in populations in which we found 0/10 colonies (that is, where it could have been rare relative to resident *E. coli* but still present), we additionally screened the DNA extracted from the community (described in the amplicon sequencing section) of the final time point by PCR (as described before for the colony PCR, but using 30 ng of DNA as template). We did this for all microcosms from the community treatment and detected PCR products in all cases in which we detected focal strain by plating and colony PCR, and none of the cases in which we did not, consistent with our analysis of individual colonies and suggesting that in those microcosms the focal strain had been completely excluded during the experiment.

To estimate total microbial abundance in each microcosm supplemented with the microbiome, we used flow cytometry. We diluted samples by 1:10,000 with phosphate-buffered saline (PBS) and stained them with SYBR Green (Invitrogen, Thermo F. Scientific). We used a Novocyte 2000R (ACEA Biosciences, San Diego, CA, United States of America), equipped with a 488-nm laser and the standard filter setup for the flow cytometric measurements.

We froze samples after every transfer from each microcosm at −80˚C, and at the end of the experiment, we isolated two sets of focal-strain colony isolates for sequencing. First, we randomly picked a single focal-strain colony isolate from each microcosm in which the focal strain was detected at the end of the experiment (from streptomycin plates; $n = 33$). Second, we randomly picked a single ampicillin-resistant colony isolate of the focal strain from each of the eight populations at the end of the experiment in which they were detected (from streptomycin + ampicillin plates; $n = 8$). We grew each colony isolate overnight in LB (with ampicillin for the ampicillin-resistant isolates), mixed 1:1 with 50% glycerol and stored at −80˚C.

## Whole-genome sequencing and bioinformatics

We sequenced all of the randomly selected ($n = 33$) and ampicillin-resistant ($n = 8$) focal-strain colony isolates (S6 Table). Prior to DNA extraction, we grew each isolate overnight in LB, then concentrated the liquid cultures by centrifugation and extracted DNA with the Magattract kit (Qiagen, Hilden, Germany) according to the manufacturer's protocol. Quality and quantity of

DNA was assessed with Nanodrop (Thermo Fisher) and Qubit (Thermo Fisher). At the Functional Genomics Center (ETH Zürich/University of Zürich), DNA was processed with Nextera library prep kit and sequenced on the Illumina Hiseq 4000 platform. We filtered raw sequencing reads with trimmomatic version 0.38 [82] and mapped the reads to the reference with snippy version 0.9.0 (https://github.com/tseemann/snippy) to detect variants. Deletions were identified by using the coverage analysis tool of CLC Genomics Workbench 11 (Qiagen) with a read threshold value of 0 and a window size of 1. We identified IS elements with ISfinder web server (database from July 2018) [83] in the ancestor genome and used these sequences to detect IS movements in the evolved strains with ISMapper version 2.0 [84].

We additionally sequenced 12 ampicillin-resistant resident *E. coli* colony isolates (not the focal strain, S6 Table). We isolated these colony isolates from microcosms filled with faecal slurry from human donors 1 and 3 at the final time point of the experiment (6 for each donor, each from a different microcosm). We picked, grew, and sequenced these colony isolates as described above for focal strain isolates. We then made de novo assemblies of the resulting sequences with spades version 3.13.0 [85] and annotated them with prokka version 1.13.7 [86]. Additionally, we sequenced one of these resident, resistant *E. coli* isolates from both human donors (1 and 3) with the Oxford nanopore long-read sequencing platform MinION at University Hospital Basel, Switzerland. These genomes were assembled using Unicycler v0.4.8 with a hybrid assembly approach combining MinION and Illumina reads. Assembly statistics can be found in S7 Table. We screened for known antibiotic-resistance genes and the presence of plasmid replicons by a local blast query against the resfinder (downloaded October 2018) [87] and plasmidfinder (downloaded October 2018) [47], which is a repository of whole-plasmid sequences from members of the Enterobacteriaceae. To identify genes that are involved in mating pair formation or mobilisation of the plasmid, we screened the genome annotation files and blasted potential candidate genes against the NCBI nucleotide database to verify them.

## Mating experiments with plasmids from resident *E. coli*

We aimed to determine whether the resident *E. coli* colony isolates could act as plasmid donors for our focal strain (transferring antibiotic-resistance plasmids via conjugation). Because the replicate *E. coli* colony isolates that we sequenced from each human donor (1 and 3) were almost identical on the DNA sequence level (S2 Table), we randomly chose one colony isolate from each human donor as a potential plasmid donor strain. We used a focal strain as the potential plasmid recipient in these experiments, which was only different from the focal strain used in the main experiment by addition of a dTomato marker and a chloramphenicol resistance cassette (enabling us to detect transconjugants by selective plating). In the first set of mating experiments, we grew overnight cultures of the potential plasmid donor strains and the potential plasmid recipient strain in LB at 37°C with shaking. We then pelleted the overnight cultures by centrifugation (3,000 rpm, 5 min), washed them twice with PBS, and resuspended them in 300 μl PBS. We then mixed the donor and recipient strains 1:1 (v:v), transferred 100 μl of this mixture to each of three replicate LB plates, and incubated the plates for 6 h at 37°C. After that, we washed the cells off the plate with 500 μl PBS and streaked out 50 μl of this plate wash on LB plates supplemented with ampicillin and chloramphenicol. To verify plasmid uptake by the transconjugants, we used a colony-PCR screen with primers targeting the recipient strain (same primer set we used above to identify the focal strain) and three additional primer sets targeting the beta-lactamase gene $bla_{Tem}$-1b (blaFW 5′-TGCAACTT-TATCCGCCTCCA-3′; blaRV 5′-TTGAGAGTTTTCGCCCCGAA-3′), the *traC* gene (traFW 5′-TCGATAAACGCCAGCTGGTT-3′; traRV 5′-AGGTGAAAACCCACAGCGAA-3′), and

the replicon *IncFIC (FII)* (IncFW 5′-CACCATCCTGCACTTACAATGC-3′; IncRV 5′-TCAGGCCCGGTTAAAAGACA-3′) with the same PCR reaction mix and settings as described above.

To test whether environmental conditions affected conjugation efficiency, we performed a second set of mating experiments in four different conditions: solid LB agar, liquid LB, liquid anaerobic LB, and anaerobic sterile faecal slurry. We prepared the liquid LB (Sigma-Aldrich) and LB agar (Sigma-Aldrich) for all treatments in the same way (independent of whether they were aerobic or anaerobic), supplementing them with 0.5 g/l L-Cysteine and 0.001 g/l Resazurin. For the anaerobic LB treatment, we transferred 0.9 ml LB to each Hungate tube, flushed the headspace with nitrogen, and sealed it with a rubber stopper before autoclaving. For the anaerobic sterile faecal slurry treatment, we added 0.45 ml LB to each Hungate tube and 0.45 ml thawed slurry under anaerobic conditions, before flushing the headspace with nitrogen, sealing, and autoclaving. Prior to each mating assay, recipient and donor strains were inoculated, washed, concentrated, and mixed exactly the same way as described above for the first set of mating experiments. For each solid and liquid treatment, four replicates were inoculated with 100 μl of the 1:1 donor recipient mix, either under aerobic or anaerobic conditions according to the respective treatment, and all tubes and plates were incubated for 6 h under static conditions at 37°C. We stopped the mating assay by either vortexing the liquid cultures or washing off the cells from the plates with 1 ml of PBS. One hundred microliters of each bacterial suspension was then plated on selective agar plates containing either chloramphenicol to count total number of recipient cells or a mix of chloramphenicol and ampicillin to count the number of transconjugants. After 24 h, we calculated transconjugant frequencies by dividing colony-forming unit (CFU) counts of plasmid-positive colonies by the total CFU count of recipient cells [69].

We measured susceptibility to ampicillin for the ancestral focal strain from the main experiment, the focal strain used in the conjugation experiment (with the dTomato tag), one focal-strain transconjugant (with the plasmid from the resident *E. coli* isolate of human donor 1), the resident *E. coli* isolate of human donor 1 (carrying the same plasmid), and all eight ampicillin-resistant focal-strain isolates from the main experiment. We did this by measuring OD600 after 24-h incubation at various concentrations of ampicillin. We prepared overnight cultures of each isolate in a randomly organised master plate and then inoculated the susceptibility assay using a pin replicator to transfer approximately 1 μl of the overnight cultures to assay plates filled with 100 μl of 0–60 μg/ml ampicillin per well. We measured OD at 0 h and after 24 h with a NanoQuant infinite M200Pro plate reader (Tecan).

## Amplicon sequencing

We thawed samples of fresh faecal slurry from 0 h and samples from each microcosm in the community treatments after 24 h and 168 h on ice and homogenised them by vortexing. We concentrated each slurry sample by centrifuging 1.5 ml of each sample at 3,000 rpm directly in the bead beating extraction tube, before removing the supernatant and repeating this step, resulting in a total volume of 3 ml of each slurry sample. We then extracted the DNA from this concentrate following the protocol of the powerlyzer powersoil kit (Qiagen). DNA yield and quality were checked by Qubit and Nanodrop.

We amplified the V3 and V4 region of the 16S rRNA gene with three slightly modified universal primers [88] with an increment of a 1-nt frameshift in each primer [89] to increase MiSeq sequencing output performance between target region and Illumina adapter. The target region was amplified by limited 17-cycle PCR with all three primer sets in one reaction for each sample. We cleaned up PCR products, and in a second PCR, adapters with the Illumina

barcodes of the Nextera XT index Kit v2 were attached. We checked 10 randomly selected samples on the Tapestation (Agilent, Basel, Switzerland) for the proper fragment size. We quantified library by qPCR with the KAPA library quantification Kit (KAPA Biosystems, Wilmington, MA, USA) on the LightCycler 480 (Roche, Basel, Switzerland). We normalised quantified samples, pooled them in equimolar amounts and loaded the library with 10% PhiX on the Illumina MiSeq platform with the Reagent Kit V3 at the Genetic Diversity Center (ETHZ).

Sequencing reads were trimmed on both ends with seqtk version 1.3 (https://github.com/lh3/seqtk), and amplicons were merged into pairs with flash version 1.2.11 [90]. USEARCH version 11.0.667 [91] was then used to trim primer sites, and amplicons were quality filtered using prinseq version 0.20.4 [92]. We clustered the quality-filtered sequences into zero-radius operational taxonomic units (ZOTUs) using USEARCH. We used Sintax implemented in the USEARCH pipeline with the SILVA database [93] for the taxonomic assignment of the ZOTU sequences. For the analysis of taxonomic data including plotting Shannnon diversity, relative proportions of taxa and to generate the PCoA plots, we used the Phyloseq package in R [94]. To estimate the frequency of *E. coli* relative to total Enterobacteriaceae, we first isolated all ZOTUs assigned to Enterobacteriaceae and divided these reads into *E. coli* and other bacteria. We blasted both groups against the SILVA database to check whether the reads were assigned to the right group.

## Statistical analyses

We used R 3.5.1 [95] for all analyses. To test whether focal-strain abundance differed between treatments, we used a generalised linear mixed-effects model with the glmmadmb function of the glmmADMB package, with zero inflation and a Poisson error distribution [96]. For this analysis, we excluded the basal medium treatment and used time, antibiotic (with/without), resident microbial community (with/without), and human donor (1/2/3) as fixed effects and replicate population as a random effect. The model was reduced by removing nonsignificant ($P > 0.05$) interactions using F-tests. $P$ values for interaction terms in the reduced model were obtained with type II Wald chi-squared tests. To test whether there was an inhibitory effect of sterilised slurry on the focal strain, we used the glmer function of the lme4 [97] R package. For this analysis, we included only community-free and antibiotic-free treatments, with focal-strain abundance as the response variable, donor as a fixed effect, replicate population as a random effect, and a Poisson error distribution. After finding interactions between the effects of resident microbiota depending on human donor and antibiotic, we analysed subsets of the dataset to look at ampicillin-free treatments only and individual human donors, using the same approach as for the main model.

To analyse the effects of ampicillin and community presence/absence on the competitive fitness of mutants and transconjugants (see S1 Methods), we used a linear model with the lmp function of the lmperm package [98]. Here we took relative fitness for the respective mutant or transconjugant as the response variable and antibiotic concentration and community presence/absence as fixed effects, testing factor effects by permutation test (accounting for the non-normal distribution of our fitness data).

To analyse differences in total bacterial abundance in the community treatments, we used a linear mixed-effects model with the lmer function of lme4 [97], with a Poisson error distribution. Time, donor, and antibiotic were fixed effects and replicate population a random effect. To analyse variation of Shannon diversity, we used a linear mixed-effects model with the lme function of nlme [99]. We excluded time point 0 h from the analysis and included time, donor, and antibiotic as fixed effects and replicate population as a random effect. To analyse similarities of microbiome samples based on the 16S rRNA data, we applied the Bray-Curtis distance

metric with the ordinate function of the Phyloseq R package to get coordinates for the PCoA. On this dataset, we ran a permutational multivariate analysis of variance (PERMANOVA) with the adonis function of vegan [100], using the distance matrix obtained from the PCoA analysis but omitting time point 0 h. We did this separately for time points 24 h and 168 h.

## Supporting information

**S1 Fig. Summary of experimental evolution in faecal slurry.** Treatments consisted of basal medium only, basal medium supplemented with sterilised faecal slurry (without the resident microbial community) from one of three human donors, or basal medium supplemented with sterilised faecal slurry to which the resident microbial community had been reintroduced (with community). After inoculation, all treatments were incubated for 2 h at 37˚C, before 7 µg/ml ampicillin was added in the antibiotic treatment. Every 24 h, we sampled each microcosm and transferred an aliquot to fresh medium (either basal medium or sterilised faecal slurry) with or without antibiotics. We serially diluted each sample and spread it on chromatic agar plates with or without antibiotics to quantify focal-strain abundance (verified by colony PCR) and to screen for resistance. We sequenced focal-strain isolates from the final time point and investigated community composition by 16S rRNA gene amplicon sequencing. We monitored total bacterial abundance in the community treatments by flow cytometry.
(TIF)

**S2 Fig. Variable similarity of microbial communities across time and treatment groups.** Each panel shows samples from a single human donor, with the same axes used in each panel. Points show the initial sample (0 h) and microcosms from 24 h and 168 h with and without antibiotics (legend at right). Similarities between communities were calculated by Bray-Curtis distance and plotted using principal coordinate analysis (see Material and methods). Data are deposited in the European Nucleotide Archive under the study accession number PRJEB33429.
(TIF)

**S3 Fig. Abundance of sequences associated with the focal strain, total *E. coli*, and the resistance plasmid from the microbiota of human donor 1, inferred with qPCR.** Each panel shows the copy number of sequences detected with primers specific for the focal strain, total *E. coli*, and the resistance plasmid (see legend; further details of primers in S1 Methods) at time point 0 h (left panel), 24 h (middle panel), and 168 h (right panel). Each point shows the mean of three technical replicates. Reactions in which no amplification was detected are shown at $10^0$. We expect plasmid copy number to reflect the abundance of plasmid donor cells, because coverage analysis of whole-genome sequencing data indicated a copy number per cell of approximately 1. For the focal strain and total *E. coli*, the copy number of sequences does not necessarily reflect the total number of cells of each type, but changes in strain abundance over time would nevertheless be expected to result in strongly correlated changes in sequence copy numbers over time. Data are deposited in the Dryad repository: https://doi.org/10.5061/dryad. t1g1jwszq [40]. qPCR, quantitative PCR.
(TIF)

**S4 Fig. Agarose gel electrophoresis picture of the PCR products specific for plasmid genes and a chromosomal marker of the focal strain.** We used these primer sets to verify plasmid uptake of the transconjugants. Primers are given in the main text in the Material and methods section.
(TIF)

**S5 Fig. Competitive fitness of transconjugants and mutants relative to the ancestral focal strain in the presence and absence of resident microbial communities with no ampicillin, sub-MIC ampicillin, and supra-MIC ampicillin.** (A) Final cell densities of competing strains (see legend; Transconjugant is a transconjugant of the focal strain carrying the plasmid from human donor 1, in the left panel; Mutant is an evolved isolate with increased ampicillin resistance from the community-free treatments with slurry from human donor 1, in the middle panel, or human donor 3, in the right panel; Ancestor is the respective ancestral focal strain). Data are shown after 24 h of competition in sterile slurry or community treatments, with and without low or high concentrations of ampicillin (x-axis). (B) Fitness of the transconjugant or mutant relative to the ancestor, calculated as the difference of their Malthusian growth rate in the same experiment. In both panels, the three points show three replicates of the experiment. Data are deposited in the Dryad repository: https://doi.org/10.5061/dryad.t1g1jwszq [40]. MIC, minimal inhibitory concentration.
(TIF)

**S6 Fig. Competitive fitness of transconjugants (carrying the plasmid from resident *E. coli* of human donor 1) relative to evolved isolates (from community-free treatments with faecal slurry from human donor 1, left, and human donor 3, right).** In each panel, relative fitness of the transconjugant strain is shown as the difference in Malthusian growth rates compared with the respective evolved isolate (see S1 Methods). Competitions were done in sterile faecal slurry or the presence of the resident microbial community and with no, low, or high ampicillin concentrations (x-axis). Each point shows a different replicate. Data are deposited in the Dryad repository: https://doi.org/10.5061/dryad.t1g1jwszq [40].
(TIF)

**S7 Fig. Abundance of the focal *E. coli* strain and resident *E. coli* strains isolated from human donors 1 and 3 (see legend) in monoculture (left) and in coculture (right).** Each strain was grown in monoculture in the absence of antibiotics, and each coculture combination was grown in the presence and absence of ampicillin (x-axis). Each point shows a different replicate. Data are deposited in the Dryad repository: https://doi.org/10.5061/dryad.t1g1jwszq [40].
(TIF)

**S8 Fig. Effect of starting bacterial density on growth inhibition by ampicillin ('inoculum effect').** Changes in bacterial abundance over 24 h are shown using three different quantification methods (OD, top panel; plating and CFU counting, middle panel; flow cytometry, bottom panel). In each panel, the change in abundance is shown for four starting densities (see legend) and at four antibiotic concentrations. In each panel, the change between 0 h and 24 h is shown (in OD in the top panel, in CFU/ml in the middle panel, and in recorded events/ml in the bottom panel). Each point shows the mean of three replicates; error bars show 1 SD. Data are deposited in the Dryad repository: https://doi.org/10.5061/dryad.t1g1jwszq [40]. CFU, colony-forming units; OD, optical density.
(TIF)

**S1 Table. Abundance of the focal *E. coli* strain in treatments with and without ampicillin after 24 h and averaged over the entire experiment.**
(PDF)

**S2 Table. Fraction of focal strain on total *E.coli* abundance determined by qPCR and a mixed calculation based of colony PCR, flow cytometry, and amplicon data.** qPCR,

quantitative PCR.
(PDF)

**S3 Table. Genomic variants found in randomly selected colony isolates of the focal strain picked from ampicillin-free agar plates at the end of the experiment.**
(PDF)

**S4 Table. Antibiotic-resistance genes, plasmid replicons, and genes involved in conjugative transfer and formation of type VI secretion system found on plasmid 1 of isolates from human donor 1 resident *E. coli* community and on the chromosome of human donor 3 resident *E. coli* isolates of each replicate population.**
(PDF)

**S5 Table. IC90 values of ancestor and ampicillin-resistant evolved strains.** IC90, concentration required to reduce growth by 90%.
(PDF)

**S6 Table. List of all sequenced isolates.**
(PDF)

**S7 Table. Assembly statistics for genome sequencing on Illumina and MinION platform of resident *E. coli* isolated from the resident microbiota of human donors 1 and 3.**
(PDF)

**S1 Model. Modelling of plasmid transfer and transconjugant growth.**
(DOCX)

**S1 Methods. Supporting materials and methods.**
(DOCX)

## Acknowledgments

We thank the Genetic Diversity Center (ETH Zürich), the Functional Genomics Center (ETH Zürich/University of Zürich), Jean-Claude Walser for sequencing and bioinformatics support, Adrian Egli and Helena Seth-Smith from University Hospital Basel for help with MinION sequencing, Jana Huisman for help with simulations, and Vera Beusch for help compiling estimates of plasmid transfer rates from literature.

## Author Contributions

**Conceptualization:** Michael Baumgartner, Angus Buckling, Alex R. Hall.

**Data curation:** Michael Baumgartner.

**Formal analysis:** Michael Baumgartner.

**Funding acquisition:** Alex R. Hall.

**Investigation:** Michael Baumgartner, Alex R. Hall.

**Methodology:** Michael Baumgartner, Florian Bayer, Katia R. Pfrunder-Cardozo, Angus Buckling, Alex R. Hall.

**Project administration:** Alex R. Hall.

**Resources:** Alex R. Hall.

**Supervision:** Alex R. Hall.

**Validation:** Michael Baumgartner, Alex R. Hall.

**Visualization:** Michael Baumgartner, Alex R. Hall.

**Writing – original draft:** Michael Baumgartner, Katia R. Pfrunder-Cardozo, Alex R. Hall.

**Writing – review & editing:** Michael Baumgartner, Angus Buckling, Alex R. Hall.

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
