## [Editor Report · Decision Letter 0]

15 Aug 2019

Dear Dr Baumgartner, 

Thank you for submitting your manuscript entitled "Resident microbial communities inhibit growth and antibiotic resistance evolution of Escherichia coli in human gut microbiome samples" for consideration as a Research Article by PLOS Biology.

Your manuscript has now been evaluated by the PLOS Biology editorial staff as well as by an academic editor with relevant expertise and I am writing to let you know that we would like to send your submission out for external peer review.

*Please be aware that, due to the voluntary nature of our reviewers and academic editors, manuscripts may be subject to delays during the holiday season. Thank you for your patience.*

Please re-submit your manuscript within two working days, i.e. by Aug 17 2019 11:59PM.

Kind regards,

Lauren A Richardson, Ph.D

Senior Editor

PLOS Biology

---

## [Decision Letter · Decision Letter 1]

18 Sep 2019

Dear Dr Baumgartner,

Thank you very much for submitting your manuscript "Resident microbial communities inhibit growth and antibiotic resistance evolution of Escherichia coli in human gut microbiome samples" for consideration as a Research Article at PLOS Biology. Your manuscript has been evaluated by the PLOS Biology editors, an Academic Editor with relevant expertise, and by independent reviewers.

The reviews of your manuscript are appended below. You will see that the reviewers find the work potentially interesting. However, based on their specific comments and following discussion with the academic editor, I regret that we cannot accept the current version of the manuscript for publication. We remain interested in your study and we would be willing to consider resubmission of a comprehensively revised version that thoroughly addresses all the reviewers' comments. We cannot make any decision about publication until we have seen the revised manuscript and your response to the reviewers' comments. Your revised manuscript would be sent for further evaluation by the reviewers.

Of particular note, the Academic Editor highlights the need to: 1) determine the role of inoculum size and/or antibiotic concentration in the outcome of resistance evolution, 2) contextualize the results of the study with the well-known concept of colonization resistance and better highlight what is known and what is new, 3) provide a better quantification of population abundances, 4) clarify, or at least further discuss, the importance of specific microbiota compositions to the outcome of resistance evolution. Due to the small sample size of fecal microbiota samples used, it is difficult to make strong general conclusions and the limitations must be made clear, as noted by Rev #3. While the Academic Editor appreciates that addressing the point raised by Rev #2 in point #2 (request for assaying temporal stability of bacteria and to explain the differences between donors) would improve the manuscript, we will not require this data in a revision.

We appreciate that these requests represent a great deal of extra work, and we are willing to relax our standard revision time to allow you six months to revise your manuscript. Please email us (plosbiology@plos.org) to discuss this if you have any questions or concerns, or think that you would need longer than this. At this stage, your manuscript remains formally under active consideration at our journal; please notify us by email if you do not wish to submit a revision and instead wish to pursue publication elsewhere, so that we may end consideration of the manuscript at PLOS Biology.

Your revisions should address the specific points made by each reviewer. Please submit a file detailing your responses to the editorial requests and a point-by-point response to all of the reviewers' comments that indicates the changes you have made to the manuscript. In addition to a clean copy of the manuscript, please upload a 'track-changes' version of your manuscript that specifies the edits made. This should be uploaded as a "Related" file type. You should also cite any additional relevant literature that has been published since the original submission and mention any additional citations in your response. 

Before you revise your manuscript, please review the following PLOS policy and formatting requirements checklist PDF: http://journals.plos.org/plosbiology/s/file?id=9411/plos-biology-formatting-checklist.pdf. It is helpful if you format your revision according to our requirements - should your paper subsequently be accepted, this will save time at the acceptance stage.

Please note that as a condition of publication PLOS' data policy (http://journals.plos.org/plosbiology/s/data-availability) requires that you make available all data used to draw the conclusions arrived at in your manuscript. If you have not already done so, you must include any data used in your manuscript either in appropriate repositories, within the body of the manuscript, or as supporting information (N.B. this includes any numerical values that were used to generate graphs, histograms etc.). For an example see here: http://www.plosbiology.org/article/info%3Adoi%2F10.1371%2Fjournal.pbio.1001908#s5.

For manuscripts submitted on or after 1st July 2019, we require the original, uncropped and minimally adjusted images supporting all blot and gel results reported in an article's figures or Supporting Information files. We will require these files before a manuscript can be accepted so please prepare them now, if you have not already uploaded them. Please carefully read our guidelines for how to prepare and upload this data: https://journals.plos.org/plosbiology/s/figures#loc-blot-and-gel-reporting-requirements.

Upon resubmission, the editors will assess your revision and if the editors and Academic Editor feel that the revised manuscript remains appropriate for the journal, we will send the manuscript for re-review. We aim to consult the same Academic Editor and reviewers for revised manuscripts but may consult others if needed.

If you still intend to submit a revised version of your manuscript, please go to https://www.editorialmanager.com/pbiology/ and log in as an Author. Click the link labelled 'Submissions Needing Revision' where you will find your submission record. 

Sincerely,

Lauren A Richardson, Ph.D

Senior Editor

PLOS Biology

Reviews

Reviewer #1: 

Baumgartner et al present new findings regarding the evolution and acquisition of antibiotic resistance as it relates to the interplay between a resident microbial community and an invading strain. The authors used an ex vivo fecal slurry culture system that allows a degree of parallelism and replicability that is hard to recapitulate with animal models. They find that the focal strain of E. coli evolves resistance to ampicillin in the absence of a microbial community but not in its presence, and that failure to mobilize plasmids bearing antibiotic resistance from endogenous E. coli strains is likely one underlying cause. 

Generally, I am interested by and like the study, and I believe that the development of ex vivo models is important to advance the mechanistic understanding of microbiome biology. The statistical analyses appear sound. However, I find parts of the study unsatisfying.

Major comments:

1. It is not very surprising to me that the evolution of antibiotic resistance depends on selection for mutations arising within a population, and that limitation of population size restricts the availability of mutants for selection to act on. This appears to be what is occurring for Donors 2 and 3 in the presence of antibiotics. The authors state “Given the variety and likely common occurrence of mechanisms that can generate such suppression of invaders (e.g. resource competition), these types of effects are likely common in species-rich communities such as the mammalian gastrointestinal tract.” If the observed phenotype in donors 2 and 3 + amp was due to resource competition, why was it not observed in the no antibiotic condition?

I can see at least two possibilities for what could be occurring for these samples. First, exposure of the focal strain to the community results in sensitization to antibiotics. Second, the focal strain, while normally able to resist inhibition by the community, is destabilized by antibiotics such that colonization resistance by the community is unmasked. 

The first possibility could potentially be addressed by taking isolates from each timepoint from these samples and performing assays such as those presented in Fig 5A, the hypothesis being that exposure to the community results in increased sensitization to antibiotics over time. The second point could potentially be addressed by asking if a pulse of antibiotics then removal results in a similar subsequent elimination of the focal strain, or if persistent addition of antibiotics is required. 

2. The plasmid from Donor 1 was able to be transferred in vitro on agar but not in liquid culture or in the static incubator. Since conjugation and mobilization of plasmids does indeed occur in vivo (for example, in Stecher et al cited in ref. 55), I am concerned that this lack of mobilization merely reflects the unsuitability of the static incubation culture system for modeling the contact-dependent interactions required for conjugation that occur in the physiological context of the gut. To this point, static incubation in a Hungate tube may not approximate the gut environment, where presumably mixing through effects of peristalsis or motility take place, and some bacteria could inhabit biofilms where horizontal gene transfer is common (De Vos 2015, NPJ Biofilms and Microbiomes; Stalder and Top 2016, NPJ Biofilms and Microbiomes). Another point relevant here is that only in Donor 1 did the focal strain decrease in abundance in the absence of antibiotics suggesting that the endogenous microbiota were particularly “good” at suppressing the growth of the focal strain. Decreased growth corresponds to fewer cells to partake in conjugative transfer (and hence perhaps the failure to recover resistant colonies), yet the data in Fig 5B suggests that the culture conditions are the primary effect.

3. I think it would be very useful to have a more granular understanding of the Enterobacteriaceae expansion at the species level (or even just at the level of total E. coli abundance). The relative abundance taxonomic data show an increase in Enterobacteriaceae in the later timepoint, to between ~25-50%. The authors state that this includes the focal E. coli strain, but at between 0-30% of the total Enterobacteriaceae. What proportion of the Enterobacteriaceae are endogenous E. coli? Is expansion of endogenous E. coli driving the overall increase across donors? This could be accomplished by qPCR analysis, for example. 

Minor comments: 

1. The amount of detail in the figure legends could be improved. An example is Fig 5A, where the assay itself is not described. Extrapolating from the methods, the authors took 1ul of an overnight culture and diluted it 1:100 in media containing antibiotic. Is the starting OD600 really ~0.75 after a 1:100 dilution from an overnight? A more traditional way to do this experiment would be to back dilute further, allow cultures to grow to a defined OD600 value (say 0.6), before adding antibiotic. This ensures that viable cell counts from exponential phase are more consistent across conditions. 

2. The plot in Figure 3 is difficult to look at (axis labels are challenging to read due to density of text). I suggest limiting the text somehow, or moving this to supplemental material.

3. The authors do not cite any of the numerous studies utilizing metagenomics to assess the mobility of antibiotic resistance genes in humans. For instance, Jiang, Alm and colleagues used metagenomic analyses to report that antibiotic resistance often spreads through mobile genetic elements in human gut samples (bioRxiv, 2017 https://doi.org/10.1101/214213).

--------------

Reviewer #2: 

The paper by Baumgartner et al. shows how the presence of a resident bacterial community can modulate the suppressive effects of an antibiotic on growth and colonization by an invader, in this case an E. coli focal strain in an experimental in-vitro system. They also compare antibiotic resistance evolution in the focal strain, in the presence and absence of the resident community, and find that in the presence of competition with a resident community, antibiotic resistance evolution was also suppressed. In fact resistance only evolved by chromosomal mutation and in the absence of a resident microbial community. These genetic mutations are described in detail. The lack of resistance evolution by plasmid-transfer in the presence of a resident community could not be explained via lack of available resistance plasmids among such bacteria, and the authors claim in the paper that it may be due to genetic constraints in one case (resident bacteria from Donor 3), and physical constraints in the other (resident bacteria from Donor 1). The paper is well-written and the experiments and statistical analyses reasonably performed, although some of the results in my view are not extremely novel or surprising, and moreover some of the empirical patterns reported are simply reported and not integrated with each other, leaving unclear what is it that one exactly learns from them. My points are listed below:

1. The main results on colonization resistance from resident microbiota are known for a number of years now, and in in-vivo settings, so the first results obtained by the authors are only incremental, testing colonization resistance in a particular case in vitro. I have a remark about the statistical analyses performed to prove this point in the paper, that all the experiments are analyzed simultaneously in the model (page 6). It is only in this way by aggregating antibiotic-free and antibiotic-treatment dynamics that colonization resistance emerges from the results of this paper. In fact, if one looks at figure 1, it is only in 1/3 of "hosts" or cases, that colonization resistance is observed in the 'baseline regime' without antibiotics. I think unifying all experiments loses the patterns in the data: a more accurate representation and modeling of this data, would be separating the antibiotic-free and antibiotic-treatment results. What is interesting, is that even when colonization resistance was not observed in the absence of antibiotics, and instead invasion and coexistence occurred, suppressive effects by residents were observed in the presence of antibiotics. Thus antibiotics modulate the effective competition between resident community and invader in donors 2 and 3, and amplify the competitive exclusion in donor 1. I think the author's general mechanical application of statistical modeling to all data in a single shot misses these intricate patterns.

2. The other results about total bacterial abundance and diversity. 

- Did the authors perform any test on the stability of total bacterial abundance in the absence of antibiotics?

- The fact that antibiotic effect was time-dependent is nothing new, it is expected and has been shown from mechanistic models of bacteria-antibiotic interaction and also experimentally (see e.g. the Regoes paper on the pharmacodynamic model 2001) so applying statistical analyses to this phenomenon brings nothing really new, only shows that such an expected pattern applies also here.

- How are the differences between Donors related? So far the paper does not make any attempt to integrate the donor variation across different metrics. Results are presented, differences are shown that exist, but no linkage between them, e.g. via some population dynamic modeling. Or are they unrelated? Just descriptive statistics of the resident community in different cases? Then the question is what did we learn? 

- A similar point applies to the changes in diversity over time: time in this dataset emerges to be a stronger predictor of changes in diversity than presence/absence of antibiotics. What is the meaning of this? The authors just report this result, but do not explain or interpret its biological consequence. This finding suggests the selection going on in the absence of antibiotics (to in-vitro experimental conditions) is stronger than the selective pressure exerted by the antibiotics. Does this have to do with the dose used, which was sub-MIC? This could be important given other literature findings that antibiotics reduce drastically microbiota diversity. Links with clinical/epidemiological findings actually are important, and should be made here.

3. The finding about antibiotic resistance evolution.

I think this is the most interesting part of the paper, and it's here that this article's strength lies: in the what/how/ and why of such patterns. Unfortunately however, I found this section somewhat incomplete. I agree that mechanisms should be studied, as the authors do, but what about population dynamics of resistance transfer? Efficient evolution via chromosomal mutation or horizontal gene transfer/plasmids depends on the population abundances and frequencies of the donor/recipient cells and corresponding process rates. The authors do not attempt to do any quantification of population dynamics here, and only focus on the molecular constraint and physical constraint, and show that indeed despite the huge beneficial effect of extant plasmids, the transfer did not happen. 

It would have been useful to have conducted the experiment at one sub-MIC dose as the authors have done, and at one supra-MIC dose, to see whether the selection pressure imposed actually changes the rate of plasmid spread. In retrospect, such resistance, even if it provided huge benefits, did not evolve because (perhaps?) it was not needed at such low selection pressure. A mathematical model would help to quantify some of these expectations. 

I also find the listing of genetic mutations (in parallel or specific) that occurred in the absence of antibiotics something that did not add much to the main point of this paper, which in my view is about interaction between resident and invader. Now why did no such chromosomal mutations evolve in the mixed cases? Is it because they have a very strong fitness cost in competition? Can this be tested? How? As these results stand now, they do not add much insights to the main point of this paper. 

5. Overall I find the paper interesting, showing four very important and unexpected outcomes, that deserve to be more central to the paper:

- no colonization resistance in the absence of antibiotics (figure 1, donors 2,3) suggesting that multiple scenarios are possible depending on donor-invader combinations - but this is not commented upon

- plasmid-mediated resistance is not always going to evolve in the invader, even if it is present in a majority background population

- competition resident community-invader amplifies the suppressive antibiotic effects on an invader.

-competition resident community-invader reduces the evolutionary potential of invader populations. 

All the other analyses in my view, are secondary and thus require less focus in the main text.

--------------

Reviewer #3: 

Baumgartner et al present an interesting study on how microbial communities may inhibit growth and antibiotic resistance evolution of a focal bacterium, in this case E. coli. This study is important for three main reasons:

(i) It is one of very few studies to provide experimental evidence that presence of the gastrointestinal microbiota changes dynamics of antibiotic resistance evolution, emphasizing the role of community interactions for evolutionary change

(ii) It is the presence of both the microbiota and an antibiotic that enhances E. coli extinction and reduces antibiotic resistance evolution, while E. coli is able to establish itself in the microbiota communities in the absence of antibiotics and to evolve antibiotic resistance in the absence of the microbiota.

(iii) Resistance evolution was not mediated by horizontal gene transfer (HGT), even though resistance-encoding mobile elements were present in the microbiota community, highlighting that HGT may be limited by genetic and environmental constraints and generally represents a rare event.

The methods are well chosen and developed; the results are reliable and have been very well analysed statistically. The study is in principle suitable for publication in Plos Biol. However, several major and minor problems still need to be improved.

Major comments:

1) The study is important because of the novel experimental approach and the insights obtained (see above points (i) and (ii)). Yet, the authors should acknowledge the study’s limitations, as only one focal strain, one antibiotic and three microbiota samples were tested. Thus, broader generalizations should be avoided and the limitations briefly described in the discussion.

2) The above point (ii) should be highlighted more clearly in the discussion. It is mentioned in the discussion (lines 283-285), yet in my opinion its importance does not become sufficiently clear. The authors found that the interaction between microbiota and antibiotics enhances E. coli extinction and minimizes resistance evolution in the surviving bacteria, whereas only one of the two does not. This is important, because it could otherwise be criticized that presence of the microbiota alone drives E. coli to extinction (which is usually not the case) or that E. coli is not able under these experimental conditions to evolve antibiotic resistance (which it can in the absence of the microbiota). 

3) The authors should assess to what extent an inoculum effect could have influenced the results. If I understood the methods correctly, then antibiotics and E. coli are used at identical concentrations in the treatment with or without microbiota communities. If correct, then the treatment with microbiota has much higher density of bacterial cells and the effective concentration of the antibiotic per bacterial cell is diluted. In turn, the focal E. coli experienced a lower effective dose of the antibiotic. In contrast, E. coli in the treatment without the microbiota experienced a much higher effective dose of the antibiotic. If correct, then it may not be surprising that E. coli only evolved antibiotic resistance in the treatment with the higher selective pressure. Please note that ampicillin is a beta-lactam antibiotic, for which such inoculum effects have been well characterized. See for example: Udekwu et al. (2009) J. Antimicrob. Chemother. 63, 745–757; or Nicoloff et al. 2019 Nature Microbiol. 4, 504–514. One option is to measure ampicillin ICs or determine ampicillin MIC for E. coli under the different cell densities.

4) Ampicillin was used at IC90, according to the methods (Line 429). Yet, there is no reduction in cell number in the microbiota-free treatments, not even at the beginning, when resistance is unlikely to have evolved de novo and/or have spread through the population. Can the authors explain this? In the worst case, some aspect of the experimental design inhibited activity of the antibiotic.

5) Genome sequencing data: The sequence data is not publicly available. Moreover, some summary statistics should be provided, including number of contigs, contig lengths, etc. Moreover, it is unclear whether duplications have contributed to resistance evolution. The duplication of genomic regions with resistance genes is well known to contribute to fast resistance emergence in E. coli. Therefore, this must be evaluated.

6) Plasmids: Without third-generation sequencing, it is difficult to assess whether the contig of interest on human donor 3 is a putative plasmid or not. It may as well be a plasmid integrated in a genomic island, and which has lost the ability to be self-transferred. Long-read sequencing would also be helpful to fully resolved the putative plasmid from human donor 1. Moreover, it does not become clear how the absence of other plasmids was excluded (see lines 239 following).

Minor comments:

1) Lines 33-35: I found the final conclusions of the abstract difficult to read. I suggest simplifying them to make them more accessible to the average reader.

2) The authors should briefly explain why ampicillin was selected for this study. 

3) Line 52: Public goods sharing could be mentioned here.

4) Line 85: ‘chromosomally’ should be changed to ‘through new mutations‘. Please note that the beta-lactamase-encoding gene may be acquired horizontally and then be integrated in the chromosome.

5) Lines 93, 117 and 133: replace ‘antibiotics‘ by ‘ampicillin’ or at least use the term in singular (only one antibiotic was used).

6) Line 97: ‘multiple’ is misleading and should be replaced by ‘three’ or ’several’.

7) Line 151: 16S should be written with a Capital S. This also applies elsewhere in the text or figures.

8) Line 152 and Figure 3: I find Supp. Fig. 2 more informative than current Fig. 3. I would at least add Supp. Fig. 2 as a sub-panel of Fig. 3 to the main text.

9) Lines 216 following: It would help the reader if information on the number of strains sequenced is provided here. Similarly, it would be useful to know how presence of plasmid genes was inferred.

10) Lines 226-228: The fact that the reads map against the single contig does not mean that these isolates share the same exact plasmid, as genomic rearrangements may have occurred. Please rephrase this.

11) Lines 250 following: It would help the reader if more precise information on MICs of the various strains would be provided, e.g. in a table.

12) Lines 287-290: The focal strain may have CRISPR-Cas or other defense mechanisms that prevent the invasion by foreign DNA, which may help to explain the troubles the authors found in transferring the plasmid by conjugation. Since the authors sequenced the strains, these mechanisms should be explored.

13) Line 312-313: The quoted papers are actually not that recent anymore. Therefore, this term should be replaced.

14) Lines 324-326: The authors do not provide any experimental evidence for the conclusion drawn. Therefore, the statement should be changed or the data provided.

15) Lines 455-456: The authors should explain the criteria used to design the primers. A quick blastn search revealed that the sequences have perfect hits with several E. coli sequences deposited in NCBI.

16) Line 462: ‘SYBR Safe’.

17) Lines 507, 512, 518-521, 605/612: Please mention the software/tool versions.

18) Line 500: Were all the of the randomly selected strains susceptible to ampicillin? This should have been tested and the data provided.

19) Line 512: ‘Ismapper’.

20) Line 556: This explanation should be moved to the first time Resazurin is mentioned (line 385).

21) Lines 586-589: Please rephrase this sentence.

22) Lines 588-589: Please provide names of the extraction kits.

23) Figure 1. Please provide information on which initial time points are shown. This cannot be inferred from the graphs.

24) Line 938 and Fig. 4 A and B: TEM in blaTEM should be subscript.

---

## [Decision Letter · Decision Letter 2]

16 Feb 2020

Dear Dr Baumgartner,

Thank you for submitting your revised Research Article entitled "Resident microbial communities inhibit growth and antibiotic resistance evolution of Escherichia coli in human gut microbiome samples" for publication in PLOS Biology. I have now obtained advice from the original reviewers and have discussed their comments with the Academic Editor. 

Based on the reviews, we will probably accept this manuscript for publication, assuming that you will modify the manuscript to address the remaining points raised by the reviewers. Please also make sure to address the data and other policy-related requests noted at the end of this email.

We expect to receive your revised manuscript within two weeks. Your revisions should address the specific points made by each reviewer. In addition to the remaining revisions and before we will be able to formally accept your manuscript and consider it "in press", we also need to ensure that your article conforms to our guidelines. A member of our team will be in touch shortly with a set of requests. As we can't proceed until these requirements are met, your swift response will help prevent delays to publication.

*Copyediting*

*Published Peer Review History*

*Early Version*

*Submitting Your Revision*

Sincerely,

Roli Roberts, PhD

Senior Editor

PLOS Biology

on behalf of

Lauren A Richardson, Ph.D, 

Senior Editor

PLOS Biology

DATA POLICY:

Regardless of the method selected, please ensure that you provide the individual numerical values that underlie the summary data displayed in the following figure panels as they are essential for readers to assess your analysis and to reproduce it: Figs 1, 2, 3AB, 5AB, S2, S3, S5AB, S6, S7, S8. NOTE: the numerical data provided should include all replicates AND the way in which the plotted mean and errors were derived (it should not present only the mean/average values).

REVIEWERS' COMMENTS:

Reviewer #1:

I am satisfied that the authors have sufficiently responded to the previous concerns raised upon review of the first submission of their manuscript. The addition of new experimental and simulation data, as well as the substantial textual revisions, have improved the clarity of the manuscript and strengthened the overall findings. 

Two minor concerns: 

- lines 309-310, refer to the T6SS as the "Type VI Secretion System". Also, "vgr" should be "vgrG"

- line 432, "type VI secretion system" should be capitalized. Also, the authors have not identified genes encoding effectors of the T6SS, just genes encoding conserved components of the apparatus. 

Reviewer #2:

After reading this revision, and having checked the authors' answers to all my previous questions, I must say they have put in a significant effort in addressing all concerns raised in the first review, and this has substantially improved their manuscript. In particular, I appreciate the competition experiments (S5 fig) and the modeling exercise S1 model, which now help to clarify the expectations for resistance evolution in this setting. The reorganization of figures and tables, and addition of new information e.g. Table S1, helps also highlight the main message of this work, which is how the presence of a microbial community microcosm impacts quantitatively the dynamics of colonization and evolution of antibiotic resistance. The authors have also commented and discussed more thoroughly in this revision the limitations of the work, which strengthens the current findings and puts them in a wider perspective.

Regarding the differences between donor samples and the condition of no resident community, I would suggest to add a line in the discussion about the importance of studying qualitative differences in terms of microbiota composition and linking those to patterns of colonization resistance and antibiotic resistance evolution. I understand that this was not possible with the current sample size (only 3 donors). But, more generally speaking, rather than presence/absence of a microbial community, as presently done in this paper, it would be an interesting challenge for the future to uncover the key (ecological) determinants in resident microbial community structure that prevent colonization by (specific) invaders and evolution of antibiotic resistance (to specific antibiotics). The authors refer to the importance of immunity and spatial structure in the discussion, but key drivers in this process are likely to be gradients in ecological structure and diversity properties of the resident microbiota itself, relative to the particular invader-antibiotic combination. And this should be mentioned. 

In my view, this study provides an important step towards empirically demonstrating colonization resistance scenarios, and highlights new protective effects of resident microbial communities in terms of preventing antibiotic resistance evolution.

Reviewer #3:

The revised manuscript is significantly improved, especially thanks to the additional data. All of my concerns have been very well addressed. Therefore, I recommend acceptance of the manuscript. As pointed out previously, this study is important because it is one of very few studies to provide experimental evidence that presence of the gastrointestinal microbiota changes dynamics of antibiotic resistance evolution, emphasizing the role of community interactions for evolutionary change. Overall, this is a manuscript very well suited for publication in PLoS Biology.

---

## [Editor Report · Decision Letter 3]

2 Apr 2020

Dear Dr Baumgartner,

On behalf of my colleagues and the Academic Editor, Isabel Gordo, I am pleased to inform you that we will be delighted to publish your Research Article in PLOS Biology. 

Early Version

PRESS 

Kind regards,

Vita Usova

Publication Assistant, 

PLOS Biology

on behalf of

Di Jiang,

Associate Editor

PLOS Biology